# Evaluating the impact of new observational constraints on P-S/IVOC emissions, multi-generation oxidation, and chamber wall losses on SOA modeling for Los Angeles, CA

Prettiny K. Ma,[1] Yunliang Zhao,[2] Allen L. Robinson,[2] David R. Worton,[3,a] Allen H. Goldstein,[3,4] Amber M. Ortega,[5,b] Jose L. Jimenez,[5] Peter Zotter,[6,c] André S. H. Prévôt,[6] Sönke Szidat,[7] and Patrick L. Hayes[1]

[1]Department of Chemistry, Université de Montréal, Montréal, QC, Canada

[2]Center for Atmospheric Particle Studies, Carnegie Mellon University, Pittsburgh, PA, USA

[3]Department of Environmental Science, Policy and Management, University of California, Berkeley, CA, USA

[4]Department of Civil and Environmental Engineering, University of California, Berkeley, CA, USA

[5]Cooperative Institute for Research in the Environmental Sciences and Dept. of Chemistry and Biochemistry, University of Colorado, Boulder, CO, USA

[6]Laboratory of Atmospheric Chemistry, Paul Scherrer Institute, Villigen, Switzerland

[7]Department of Chemistry and Biochemistry & Oeschger Centre for Climate Change, University of Bern, Bern, Switzerland

[a]now at: National Physical Laboratory, Hampton Rd, Teddington, Middlesex, UK

[b]now at: Air Pollution Control Division, Colorado Department of Public Health and Environment, Denver, CO, USA

[c]now at: Lucerne University of Applied Sciences and Arts, School of Engineering and Architecture, Bioenergy Research, Technikumstrasse 21, CH-6048 Horw, Switzerland

*Correspondence to:* Patrick L. Hayes (patrick.hayes@umontreal.ca)

## ABSTRACT

Secondary Organic Aerosols (SOA) are important contributors to fine PM mass in polluted regions, and their modeling remains poorly constrained. A box model is developed that uses recently published literature parameterizations and data sets to better constrain and evaluate the formation pathways and precursors of urban SOA during the CalNex 2010 campaign in Los Angeles. When using the measurements of IVOCs reported in Zhao et al. (2014) and of SVOCs reported in Worton et al. (2014) the model is biased high at longer photochemical ages whereas at shorter photochemical ages it is biased low, if the yields for VOC oxidation are not updated. The parameterizations using an updated version of the yields, which takes into account the effect of gas phase wall-losses in environmental chambers, show model/measurement agreement at longer photochemical ages, even though some low bias at short photochemical ages still remains. Furthermore, the fossil/non-fossil carbon split of urban SOA simulated by the model is consistent with measurements at the Pasadena ground site.

Multi-generation oxidation mechanisms are often employed in SOA models to increase the SOA yields derived from environmental chamber experiments in order to obtain better model/measurement agreement. However, there are many uncertainties associated with these ''aging'' mechanisms. Thus, SOA formation in the model is compared against data from an oxidation flow reactor (OFR) in order to constrain SOA formation at longer photochemical ages than observed in urban air. The model predicts similar SOA mass at short to moderate photochemical ages when the ''aging'' mechanisms or the updated version of the yields for VOC oxidation are implemented. The latter case though has SOA formation rates that are more consistent with observations from the OFR. Aging mechanisms may still play an important role in SOA chemistry, but the additional mass formed by functionalization reactions during aging would need to be offset by gas-phase fragmentation of SVOCs.

All the model cases evaluated in this work have a large majority of the urban SOA (70 − 83 %) at Pasadena coming from the oxidation of P-SVOCs and P-IVOCs. The importance of these two types of precursors is further supported by analyzing the percentage of SOA formed at long photochemical ages (1.5 days) as a function of the precursor rate constant. The P-SVOCs and P-IVOCs have rate constants that are similar to highly reactive VOCs that have been previously found to strongly correlate with SOA formation potential measured by the OFR.

Finally, the volatility distribution of the total organic mass (gas and particle phase) in the model is compared against measurements. The total SVOC mass simulated is similar to the measurements, but there are important differences in the measured and modeled volatility distributions. A likely reason for the difference is the lack of particle-phase reactions in the model that can oligomerize and/or continue to oxidize organic compounds even after they partition to the particle phase.

# 1. INTRODUCTION

Atmospheric aerosols are important climate forcing agents (Christensen et al., 2013), negatively impact human health (Dockery and Pope, 1994) and reduce visibility by scattering and absorbing light (Watson, 2002). However, predicting quantitatively the composition and concentrations of aerosols is challenging, in part because of their complex composition and the variety of emission sources and chemical pathways that contribute to aerosol loadings in the atmosphere (Heald et al., 2011; Spracklen et al., 2011). Atmospheric aerosols are composed of black carbon, inorganic, and organic matter, and the latter is a mixture of hundreds to thousands of compounds (Gentner et al., 2012).

Due to this complexity, organic aerosol is often categorized into two groups. Primary organic aerosol (POA) is directly emitted into the atmosphere from sources such as motor vehicles, food cooking, and biomass burning (Hallquist et al., 2009). On the other hand, secondary organic aerosol (SOA) is the product of diverse chemical reactions occurring in the atmosphere that transform more-volatile precursors such as volatile organic compounds (VOCs) into lower volatility products that are either incorporated into existing particles or form new particles. Many previous studies have shown that SOA is an important fraction of OA globally often representing more than half the total OA concentration (Zhang et al., 2007; Jimenez et al., 2009).

In SOA parameterizations for use in regional and global models, a semi-empirical approach is used in which VOCs, often the only SOA precursors considered, react with OH radicals and other oxidants to form secondary products with lower volatility at a given mass yield. These secondary semi-volatile organic compounds (SVOCs) can partition to the particle phase to form SOA (Pankow, 1994; Odum et al., 1996; Donahue et al., 2006). The parameters used in the models for the VOCs, such as the yields and product volatilities, are often determined from published chambers studies (e.g. Kroll et al., 2006; Chan et al., 2009; Hallquist et al., 2009; Presto et al., 2010). Over the past decade a number of studies have shown that traditional models that consider only the oxidation of VOCs alone predict SOA concentrations much lower than those observed in polluted urban regions (Volkamer et al., 2006; Dzepina et al., 2009; Hodzic and Jimenez, 2011; Hayes et al., 2015). As a result, several updates have been proposed in the literature to improve SOA models including new pathways for SOA formation, new SOA precursors, and increased yields for known precursors (e.g. Ng et al., 2007; Robinson et al., 2007; Ervens and Volkamer, 2010).

The volatility basis-set (VBS) approach (Donahue et al., 2006) has been used in most recent parameterizations of SOA yields. In this approach, the organic mass is distributed in logarithmically spaced volatility bins, and the SOA forming reactions then redistribute the mass from precursors such as anthropogenic and biogenic VOCs, into

bins with generally lower volatility (except for fragmentation reactions) leading to increased OA concentrations (Robinson et al., 2007; Tsimpidi et al., 2010). While the VBS provides a valuable conceptual framework for SOA modeling, substantial uncertainties remain in the correct parameters for different precursors and conditions.

In this paper we focus on investigating three interrelated questions that are responsible for important uncertainties in urban SOA modeling. The first is how to best incorporate SOA from primary semi- and intermediate volatility compounds (P-S/IVOCs), two recently-proposed types of SOA precursors. While there is now ample evidence that P-S/IVOCs are important contributors to SOA (Robinson et al., 2007; Zhao et al., 2014; Dunmore et al., 2015; Ots et al., 2016), the emissions of these precursors as well as the parameters that govern their oxidation and SOA formation are not well constrained. Also, it is well known that models of SOA that incorporate P-S/IVOCs often do not agree with measurements across a range of photochemical ages, although the modeled SOA mass varies substantially with the parameterization used (Dzepina et al., 2009; Hayes et al., 2015; Fountoukis et al., 2016; Woody et al., 2016). The second question is whether losses of semi-volatile gases to the walls of environmental chambers (Matsunaga and Ziemann, 2010; Krechmer et al., 2016) have resulted in low biases for the yields of some or all precursors, especially VOCs, as has been recently reported (Zhang et al., 2014). The third question is the appropriateness of including "aging" mechanisms in the VBS parameterization of SOA from VOCs, in which the initial oxidation reaction is followed by subsequent oxidation reactions of the first and later generation products, with each reaction resulting in a reduction of the organic volatility by, for example, an order of magnitude. These "aging" mechanisms increase VOC yields to levels much higher than those observed in chamber studies since it was perceived that the yields may be too low in chambers compared to the real atmosphere. The "aging" mechanisms were added to chamber yields that were obtained without using aging as part of the fits of the chamber data. In some model applications they improve model agreement with field measurements (Ahmadov et al., 2012), while at long photochemical ages they lead to model SOA formation that is substantially larger than observed (e.g. Dzepina et al., 2011; Hayes et al., 2015). While the inclusion of some of these new SOA precursors, updated yields, and aging can provide in some cases better agreement with measurements, the relative amount of SOA formed from VOCs (V-SOA), P-IVOCs (I-SOA), and P-SVOCs (S-SOA) is highly uncertain, and changes strongly depending on which of the above updates are implemented in a specific model. In addition, the fact that different subsets and variants of these updates can allow specific models to match SOA measurements raises important questions regarding whether or not the model mechanisms are representative of actual SOA forming processes in the atmosphere.

The notation used when discussing SOA precursors in this paper is similar to
Hayes et al. (2015). We differentiate VOCs, IVOCs and SVOCs by their effective
saturation concentration (c*). Therefore, SVOCs and IVOCs have volatilities ranging
from $c* = 10^{-2}$ to $10^2$ and $10^3$ to $10^6$ µg m$^{-3}$ respectively, while VOCs are in the bins of c*
$\geq 10^7$ µg m$^{-3}$.

Recently, we evaluated three parameterizations for the formation of S-SOA and I-
SOA using a constrained 0-D box model that represents the South Coast Air Basin during
the California Research at the Nexus of Air Quality and Climate Change (CalNex)
campaign (Hayes et al., 2015). Box models are often used to compare with ambient
measurements, and have been shown to be of similar usefulness or even superior to 3-D
models if the emissions and atmospheric transport affecting a given case study are well
constrained, and if the use of ratios to tracers can be used to approximately account for
dispersion (e.g. Volkamer et al., 2006; Dzepina et al., 2009; Hayes et al., 2015; Yuan et
al., 2015). A box model allows the evaluation of multiple model parameterizations either
previously proposed in the literature or developed from recent field and laboratory data
sets, as well as the performance of sensitivity studies, all of which would be difficult to
carry-out in more computationally demanding gridded 3-D models. There are six model
cases presented in this paper that are described in further detail below. Given the number
of model cases (including three additional model cases from Hayes et al. (2015)), it
would be very computationally expensive to use a 3-D model to evaluate all the cases.

Moreover, there are important limitations to traditional comparisons of 3-D
models' predicted concentrations against measurements, as for example discussed for the
Pasadena ground site in Woody et al. (2016). In that study, the SOA predicted by the
Community Multiscale Air Quality (CMAQ) model with a VBS treatment of OA is a
factor of 5.4 lower than the measurements during the midday peak in SOA
concentrations. This underestimation was attributed to several different factors. First, the
model photochemical age for the site was too low by a factor of 1.5. In the box model
presented in this current work, that problem is eliminated as the photochemical aging of
the urban emissions in the model is instead determined from the measured ratio of 1,2,4-
trimethylbenzene to benzene as described previously (Parrish et al., 2007; Hayes et al.,
2013). Second, it is difficult to distinguish errors due to model dispersion from those due
to emission inventories and photochemical age. Woody et al. (2016) conclude that
excessive dispersion or low emissions account for an error of about a factor of 2. Those
errors are also eliminated by the use of emission ratios in this work. After those errors are
accounted for, by analyzing the 3-D model output using similar techniques as in our box
model, the real under-prediction of SOA formation efficiency by a factor of 1.8 emerged,
compared to the initial value of 5.4 from the concentration comparisons. These errors (of
approximately 300%) in the interpretation of 3-D model comparisons, which are ignored
in most 3-D model studies, are far larger than the uncertainties due to emission ratios or
dispersion in our box model (about 10 - 20%), as demonstrated in section 2.4.
In addition, there are uncertainties in the P-S/IVOC emissions inventories used in
3-D models and in the methods used to estimate P-S/IVOC emissions from the traditional
POA inventories. In our box model, as described in further detail below, we incorporated
recently published field measurements of P-S/IVOCs to better constrain the concentration
of these species. Thus, while 3-D models are essential for simulating spatially and
temporally complex environments under the influence of many sources, in cases where
transport is relatively simple and there is a well-defined urban plume such in Pasadena
during the CalNex campaign, the box model is a valuable complementary or even
superior approach that is less susceptible to the convoluted uncertainties in 3-D models
discussed above. Another reason to use a box model is that it allows a direct comparison
against OFR measurements taken in the field (Ortega et al., 2016). The OFR provided
(every 20 minutes at the Pasadena ground site) a measure of SOA formation potential for
a photochemical age of up to two weeks. To the best of our knowledge, 3-D models have
not yet been adapted for comparison against OFR data. Finally, box models are more
widely usable by experimental groups (such as ours) due to reduced complexity, while
3-D models are almost exclusively used by modeling-only groups, who tend to be more
distant from the availability, use, and interpretation of experimental constraints. Thus the
use of a range of models by a range of different groups is highly beneficial to scientific
progress.
The results obtained in our previous work (Hayes et al., 2015) using a box model
indicated that different combinations of parameterizations could reproduce the total SOA
equally well even though the amounts of V-SOA, I-SOA, and S-SOA were very different.
In addition, the model over-predicted SOA formed at longer photochemical ages ($\approx$ 3
days) when compared to observations downwind of multiple urban sites. This
discrepancy suggests that the ratio of P-S/IVOCs-to-POA may have been too high in the
parameterizations evaluated. Also, as mentioned previously and discussed in Hayes et al.
(2015), the implementation of aging for VOC products remains uncertain.
The goal of this study is to use several recently published results to better evaluate
and constrain the box model introduced in our previous work, and thus facilitate the
identification of parameterizations that can be eventually incorporated into 3-D air
quality models to accurately predict SOA for the right reasons. It is important to note that
parameterizations used in the box model are based on several published measurements
taken from laboratory experiments and field studies that provide more realistic
constraints than in previous versions and that were not available to be implemented in
Hayes et al. (2015). In particular, our work here improves the box model by incorporating
recently published measurements of P-IVOCs and P-SVOCs that allow better
constraining of the concentration, reactivity, yields, and volatility of these precursors

(Worton et al., 2014; Zhao et al., 2014). In addition, given that experiments in environmental chambers may underestimate SOA yields for the VOCs due to losses of semi-volatile gases to the chamber walls (Zhang et al., 2014), the SOA yields from VOCs have been re-estimated using a very recent parameterization of these wall-losses (Krechmer et al., 2016). The wall-loss corrected yields obtained are then used in the model in a sensitivity study to evaluate the corresponding change in the modeled SOA concentrations. The model is modified based on these literature constraints. No model tuning is performed with the goal of improving the agreement with the observations. The results obtained from the new box model are compared against ambient ground site and airborne measurements, and also against recently-published oxidation flow reactor (OFR) measurements (Ortega et al., 2016). This combination of data sets allows the model to be evaluated for photochemical ages ranging up to 3 equivalent days (at $1.5 \times 10^6$ molec OH $cm^{-3}$) providing a means to evaluate the aging mechanisms of the VOCs in the VBS.

## 2. EXPERIMENTAL SECTION

## 2.1 Measurement and sampling site

The box model is constructed in order to represent the South Coast Air Basin during CalNex in spring/summer 2010. The measurements of aerosols used in this study were conducted in Pasadena, California (34.1406° N 118.1224° W), located to the northeast of downtown Los Angeles (Hayes et al., 2015). An overview of CalNex has been published previously (Ryerson et al., 2013). The location and the meteorology of the ground site at Pasadena are described in further detail in Hayes et al. (2013). Pasadena is a receptor site for pollution due to winds that transport emissions from the Ports of Los Angeles and Long Beach and downtown Los Angeles. Airborne measurements of aerosols were also carried out in the South Coast Air Basin as part of the CalNex project. A detailed description of the airborne measurements is given in Bahreini et al. (2012). Furthermore, measurements of POA composition and volatility taken at the Caldecott Tunnel in the San Francisco Bay Area reported in previous work (Worton et al., 2014) are also used to constrain the model as described below. The tunnel air samples were collected during July 2010.

Two additional datasets are used to evaluate the model. In addition to sampling ambient air, an aerosol mass spectrometer (AMS) sampled air that had been photochemically aged using an oxidation flow reactor (OFR) (Ortega et al., 2016). The OFR exposed ambient air to varying concentrations OH radicals in order to obtain photochemical ages much higher than the ambient levels observed at the Pasadena site, and the amount of SOA produced was quantified as a function of OH exposure. Moreover, radiocarbon ($^{14}$C) analysis has been performed on filter samples and results were combined with positive matrix factorization (PMF) data to determine fossil and

non-fossil fractions of the SOA components as reported in Zotter et al. (2014). The [14]C
results are used for subsequent comparison against the box model from which fossil and
non-fossil SOA mass can be estimated.

## 2.2 Model set-up

The SOA model is set-up to include 3 types of precursors: VOCs, P-IVOCs, and
P-SVOCs. The parameters used in the box model to simulate the formation of SOA from
these precursors are listed in Tables S1 to S3 of the supporting information. The box
model dynamically calculates the evolution of organic species in an air parcel as it
undergoes photochemical aging, hence producing SOA. The total SOA also includes
background SOA (BG-SOA) at a constant concentration of 2.1 $\mu$g m$^{-3}$, as determined in
our previous work (Hayes et al., 2015). The model accounts for P-SVOC emissions from
vehicular exhaust and cooking and treats POA as semi-volatile (Robinson et al., 2007). It
should be noted that the model uses CO and NO$_x$ as inputs to constrain the model, and the
SOA yields for high-NO$_X$ conditions are used, based on our previous work (Hayes et al.,
2013; 2015). Therefore, to verify model performance both predictions of VOC and POA
concentrations have been compared against field measurements and the model
performance appears to be satisfactory (Hayes et al., 2015).
A schematic of the model is shown in Figure 1. All the model cases are listed in
Table 1, and all the parameterizations are shown schematically in Figure 2. The first
model case (ROB + TSI) incorporates the Robinson et al. (2007) parameterization for
SOA formation that models P-IVOCs and P-SVOCs (i.e. P-S/IVOCs) using a single
volatility distribution and oxidation rate constant. The ROB + TSI case also uses the
Tsimpidi et al. (2010) parameterization for SOA formation from VOCs. A detailed
description of the parameters used in ROB + TSI can be found in Hayes et al. (2015), and
the ROB + TSI model case used here is identical to the case of the same name used in
that paper. Briefly, as displayed in Fig. 2A, the Tsimpidi et al. (2010) parameterization
proposes that the VOCs undergo an initial oxidation step that will form four lumped
products with different volatilities (c* = 1, 10$^1$, 10$^2$, 10$^3$ $\mu$g m$^{-3}$, where c* is the effective
saturation concentration). The first-generation oxidation products can be further oxidized,
decreasing their volatility by one order of magnitude (i.e. aging). This "bin-hopping"
mechanism repeats until the lowest volatility product is reached (c* = 10$^{-1}$ $\mu$g m$^{-3}$ in this
study and 1 $\mu$g m$^{-3}$ in other studies such as Tsimpidi et al. (2010) and Hayes et al. (2015).
The Robinson et al. (2007) parameterization proposes that the P-S/IVOCs are initially
distributed in logarithmically spaced volatility bins ranging from c* = 10$^{-2}$ to 10$^6$ $\mu$g m$^{-3}$.
Thereafter, the oxidation of P-S/IVOCs decreases their volatility by one order of
magnitude until the lowest volatility product is reached (c* = 10$^{-2}$ $\mu$g m$^{-3}$). The lowest
volatility product possible is not the same for the oxidation of VOCs versus the oxidation
of the P-S/IVOCs (10$^{-1}$ vs. 10$^{-2}$ $\mu$g m$^{-3}$, respectively). However, whether the mass is

distributed into either bin has a negligible effect on the SOA mass simulated in the box model because of the relatively high SOA concentrations during the case study.

In this work, 5 model parameterizations are tested that incorporate new measurements of IVOCs and P-SVOC volatility as well as updated VOC yields that account for wall-losses of vapors (Zhang et al., 2014; Krechmer et al., 2016). For the first new case (ROB + ZHAO + TSI), we incorporate IVOC data measured in Pasadena during the CalNex campaign as reported from Zhao et al. (2014). In particular, the measured concentrations of speciated and unspeciated IVOCs and their estimated volatility are used to constrain the initial concentration of these species (as discussed in Section 2.2.2 below) as well as to estimate their yields (Zhao et al., 2014). Therefore, we replace the inferred concentrations of IVOCs that were used in our previous work and based on the volatility distribution of Robinson et al. (2007) with concentrations that are directly constrained by measurements. In the ROB + ZHAO + TSI case the SOA formation parameters used (e.g. yields, oxidation rate constants) are taken from Zhao et al. (2014) for the IVOCs and from Hayes et al. (2015) for the VOCs and SVOCs. Hodzic et al. (2016) have also estimated the IVOC yields while accounting for wall-losses using recent laboratory studies. However, the yields reported in that study are for a single lumped species, whereas in our work we estimate the yields using 40 IVOC categories, each representing a single compound or a group of compounds of similar structure and volatility. This method allows a more precise representation of IVOC yields and rate constants in the SOA model.

For the second new case (WOR + ZHAO + TSI), the volatility distribution of P-SVOCs is updated using measurements of POA performed at the Caldecott tunnel in the California Bay Area (Worton et al., 2014). In the previous two cases described above, the relative volatility distribution of P-SVOCs was taken from the work of Robinson et al. (2007). In this distribution, the relative concentration of SVOCs increases monotonically between the $c* = 10^{-2}$ and $10^2$ µg m$^{-3}$ bins. The P-SVOC volatility distribution in the WOR + ZHAO + TSI case increases monotonically as well, but the relative concentrations in each bin are different and notably there is a much higher relative concentration of SVOCs in the $c* = 10^2$ µg m$^{-3}$ bin (see Fig. 2 and Table S3 in the supporting information). In this model case, the updated P-SVOC volatility distribution is only applied to vehicular P-S/IVOCs whereas the volatility distribution proposed by Robinson et al. (2007) is still used for cooking emissions.

Several recently published papers have found that chamber experiments may underestimate SOA yields due to the loss of semi-volatile vapors to chamber walls (Matsunaga and Ziemann, 2010; Zhang et al., 2014; Krechmer et al., 2016). A sensitivity study has been performed to explore this uncertainty by running the three model cases described above (ROB + TSI, ROB + ZHAO + TSI, and WOR + ZHAO + TSI) with a revised version of the SOA yields for VOCs that accounts for these wall losses. A

detailed description of how these updated yields were estimated is provided in the supporting information and the values can be found in Table S4. Briefly, equilibrium partitioning is assumed to hold for the organic mass found in the gas phase, particle phase, or chamber walls. The SOA yields are then obtained by refitting SOA chamber yield curves using a model that accounts for partitioning between the three compartments (particle, gas, and wall) and incorporates the equivalent wall mass concentrations published in Krechmer et al. (2016), which are volatility dependent. The SOA chamber yield curves that were refitted were first calculated using the parameters published in Tsimpidi et al. (2010). There are limits to the assumption that partitioning between the three phases occurs on short enough timescales for all four VOC product volatilities that equilibrium is reached during an SOA chamber study. Specifically, at lower volatilities ($c^* \leq 1$ µg m$^{-3}$), the partitioning kinetics of the organic mass from the particles to the chamber walls have an effective timescale of more than an hour, which is similar or longer than typical chamber experiments (Ye et al., 2016). The limiting step in the partitioning kinetics is evaporation of SVOCs from the particles to the gas phase, and therefore the exact rate of evaporation depends on the OA concentration in the chamber.

Furthermore, as described in the supporting information, the updated SOA yields for VOC oxidation result in distribution of SVOC mass into lower volatility bins compared to the original parameterization, although the sum for the SVOC yields ($\alpha_i$) remains similar. In the absence of aging, the SOA yields, Y, resulting from the wall-loss correction should be considered upper limits (MA parameterization), whereas the original yields serve as lower limits due to the considerations discussed above (TSI parameterization without aging). As shown in the supporting information (Figures S1 - S7) when aging (TSI parameterization with aging) is included the SOA yields increase beyond those observed when applying the wall loss correction for most of the VOC classes at longer photochemical ages. (It should be noted that SOA masses in Fig. S1 - S7 were calculated using the same background as for the other model cases, 2.1 µg m$^{-3}$.) This feature of the aging parameterization is likely to blame for SOA over-predictions observed at long aging times when comparing with ambient data (e.g. Dzepina et al., 2009; Hayes et al., 2015).

According to Krechmer et al. (2016) and other chamber experiments (Matsunaga and Ziemann, 2010), the gas-wall equilibrium timescale doesn't vary strongly with the chamber size. The timescale for gas-wall equilibrium reported in these previous studies was 7 - 13 minutes. Similar timescales have been calculated for a variety of environmental chambers, including chambers that were used to determine many of the yields used in this paper. In addition, Matsunaga and Ziemann found that partitioning was nearly independent of chamber treatment, reversible, and obeyed Henry's law. Thus, the effective wall concentrations determined from the chamber experiments reported in Krechmer et al. (2016) are likely applicable to other chambers with different sizes.

The three model cases accounting for wall losses of organic vapors are named
ROB + MA, ROB + ZHAO + MA, and WOR + ZHAO + MA. For these cases, the aging
of the secondary SVOCs formed from the oxidation of VOCs was not included, since
multi-generation oxidation is not well-constrained using data from chamber studies that
are run over relatively short time-scales (i.e. hours). In addition, aging and correcting for
wall-losses of organic vapors have been separately proposed to close the gap between
observed and predicted SOA concentration from pre-2007 models, and are thought to
represent the same "missing SOA mass." Therefore, we run the model with one of these
options at a time, as they are conceptually different representations of the same
phenomenology. The aging of secondary SVOCs formed from the oxidation of P-IVOCs
(and P-SVOCs) has been kept for all of the MA cases, however. To our knowledge, P-
IVOC and P-SVOC mechanisms proposed in the literature have always included aging. A
similar approach for correcting the yields as described above cannot be applied to P-
IVOCs because organics with low volatilities ($c* < 10$ µg m$^{-3}$) will partition to chamber
walls very slowly, and SVOCs from P-IVOC oxidation tend to have lower volatilities
than the SVOCs formed from VOC oxidation (Tables S1 and S2). Indeed, when trying to
refit the VOC and IVOC yield curves, the model assuming equilibrium partitioning
between particles, the gas phase, and the walls was able to reproduce the yield curves for
VOCs, but not for IVOCs. This difference in the results is consistent with equilibrium not
having been reached during the chamber studies on the IVOCs, which produce a greater
amount of lower volatility SVOCs when compared to VOCs during oxidation. These
lower volatility SVOCs have relatively slow evaporation rates from the particles, which
prevents the chamber system from reaching equilibrium (Ye et al., 2016).
Simulations of O:C have been previously evaluated in Hayes et al. (2015) using
laboratory and field data from CalNex to constrain the predicted O:C. It was concluded in
that work that it was not possible to identify one parameterization that performed better
than the other parameterizations evaluated, because of the lack of constraints on the
different parameters used (e.g. oxidation rate constant, oxygen mass in the initial
generation of products and that added in later oxidation generations, SOA yields, and
emissions). Therefore, incorporating O:C predictions into the current box model and
using those results in the evaluation discussed here would not provide useful additional
constraints.

## 2.2.1 IVOC oxidation parameterizations

An important difference between the ROB + TSI and ROB + MA cases and the
other four cases that have been updated with the IVOC measurements of Zhao et al.
(2014) is that in the ZHAO cases, the first generation of IVOC oxidation distributes part
of the product mass into four different volatility bins ($c* = 10^{-1}$, 1, $10^{1}$, $10^{2}$ µg m$^{-3}$) as is
displayed in Fig. 2E. This IVOC oxidation scheme is similar to that used for the first step
of VOC oxidation (Tsimpidi et al., 2010) as displayed in Fig. 2A and D, and has been
used to model chamber measurements of SOA from IVOCs (Presto et al., 2010).
Contrastingly, in the ROB + TSI and ROB + MA cases, a "bin-hopping" approach is used
for all P-S/IVOCs where oxidation lowers volatility by only one order of magnitude (see
Fig. 2B and C). The Robinson et al. (2007) parameters are still used for the formation of
SOA from P-SVOCs in the ROB + ZHAO + TSI and ROB + ZHAO + MA cases, but the
parameters are only applied to primary emissions in c* bins between $10^{-2}$ and $10^{2}$ μg m$^{-3}$
inclusive (i.e. the volatilities corresponding to P-SVOCs).

## 2.2.2. Determination of initial precursor concentrations

In the ROB + TSI and ROB + MA cases, the initial concentration of P-S/IVOCs is
estimated as follows. The volatility distribution determined by Robinson et al. (2007) is
assumed to represent all P-S/IVOCs emitted (Dzepina et al., 2009). The total
concentration of P-S/IVOCs is then set so that the amount of P-S/IVOCs in the particle
phase is equal to the initial POA concentration. The initial POA concentration is
determined from the product of the background-subtracted CO concentration and the
ΔPOA/ΔCO emission ratio (Hayes et al., 2015). While this ratio may change due to
evaporation/condensation or photochemical oxidation of POA, our previous work (Hayes
et al., 2013) has shown that ΔPOA/ΔCO does not change significantly at the Pasadena
ground site with observed photochemical age indicating that the ratio is insensitive to the
extent of photochemical oxidation. Furthermore, it was calculated that the ratio would
increase by 28% for an increase of OA concentration from 5 to 15 μg m$^{-3}$, concentrations
that are representative of this study. This possible source of error is substantially smaller
than current errors suggested for P-S/IVOC emission inventories in 3-D models, where
current schemes are based on scaling POA emission inventories with scaling factors that
are not well constrained (Woody et al., 2016). The same method is used for the other four
model cases, but only the initial concentration of P-SVOCs is estimated by this method
and the initial concentration of P-IVOCs is estimated separately as described in the next
paragraph. In addition, in the WOR + ZHAO + TSI and WOR + ZHAO + MA cases the
volatility distribution of vehicular P-SVOCs reported in Worton et al. (2014) is used for
estimating the initial concentration of vehicular P-SVOCs whereas the volatility
distribution of Robinson et al. (2007) is used for estimating the initial concentration of
cooking P-SVOCs.
It should be noted that the tunnel measurements do not include emissions due to
cold starts of vehicles. In the box model, only the relative volatility distribution of
vehicular POA measured during the tunnel study is used, and thus this potential source of
error does not apply to the total amount of vehicular POA emissions in the model.
However, it is still possible that the volatility distribution of POA is different during cold-
starts compared to that of POA emitted from warm-running engines. To our knowledge,
measurements of the volatility distribution of POA during cold-starts are not available at
this time. By comparing the SOA model results using two different POA volatility
distributions (Robinson et al., 2007; Worton et al., 2014), we can evaluate to a certain
extent the sensitivity of the simulated SOA concentration to the initial POA volatility
distribution.
The initial concentrations of VOCs and IVOCs are calculated by multiplying the
background-subtracted CO concentrations measured at Pasadena by the emission ratios
$\Delta VOC/\Delta CO$ or $\Delta IVOC/\Delta CO$. In the ROB + TSI and ROB + MA cases this method is
only applied to the VOCs. The initialization method for the concentrations of the VOCs
is the same for all six cases in this paper. For the biogenic VOCs, we follow the same
method as Hayes et al. (2015) to determine the initial concentrations since these
compounds are not co-emitted with CO. The emission ratios are taken from the literature
when available (Warneke et al., 2007; Borbon et al., 2013). For most of the IVOCs and
some VOCs, emission ratios are not available in the literature. The ratios are instead
determined by performing linear regression analyses on scatter plots of the IVOC or
VOC and CO concentrations measured in Pasadena between 00:00-06:00 local time when
the amount of photochemical aging was very low. During the regression analyses the x-
intercept was fixed at 105 ppbv CO to account for the background concentration of CO
determined in our previous work (Hayes et al., 2013). Thus, the slope of the resulting line
corresponds to the estimated emission ratio ($\Delta IVOC/\Delta CO$).
It should be noted that the use of VOC emission ratios to CO to estimate VOC
emissions does not assume that VOCs are always co-emitted with CO. Rather, it assumes
that VOC emission sources are individually small and finely dispersed in an urban area,
so that they are spatially intermingled with the sources of CO. Moreover, previous studies
have measured the emission ratios of anthropogenic VOCs with respect to CO and the
results show that vehicle exhaust is a major source of VOC and CO (Warneke et al.,
2007; Borbon et al., 2013). Furthermore, the ratios are consistent both temporally and
spatially. Thus, when thinking of the entire urban area as a source, the use of emission
ratios to CO is justified. As shown in Hayes et al. (2015) in the supporting information,
the modeled VOC concentrations are consistent with the measurements indicating that
major VOCs sources have not been omitted, and the smooth time variations of the VOC
concentrations support the use of a "global urban source".
**2.3 SOA model**
The VOC yields are taken from Tsimpidi et al. (2010) or determined in this work
as described below. The estimation of the IVOC yields (based on values taken from
Presto et al. (2010) and of the OH reaction rate constants for IVOCs follows the same
approach used by Zhao et al. (2014). However, instead of using the total SOA yield, *Y,*
for a fixed OA concentration as reported in Zhao et al. (2014), we use the SVOC yield, *α,*
of each c* bin. It is important to note here that the SOA yields taken from Tsimpidi et al.
and Presto et al. use a four-product basis set with $c* = 10^0, 10^1, 10^2, 10^3$ μg m$^{-3}$ and $c* =$
$10^{-1}, 10^0, 10^1, 10^2$ μg m$^{-3}$ respectively. For this box model, it is more appropriate to have
a uniform VBS in terms of the bin range utilised so a bin with a lower volatility ($c* = 10^{-1}$
μg m$^{-3}$) has been added to the VBS distribution of Tsimpidi et al. (2010). The yield for
bin $c* = 10^{-1}$ μg m$^{-3}$ is 0 for VOC oxidation, but when aging occurs mass can be
transferred into this bin. However, the change in the total V-SOA mass is negligible
because for both bin $c* = 10^{-1}$ and $10^0$ μg m$^{-3}$ the secondary products almost completely
partition to the particle phase.

The OH reaction rate constants are taken from the literature (Atkinson and Arey,
2003; Carter, 2010) as described previously in Hayes et al. (2015). During aging, the
oxidation products undergo subsequent reactions with OH radicals with a reaction rate
constant of $1 \times 10^{-11}$ cm$^3$ molec$^{-1}$ s$^{-1}$ and $4 \times 10^{-11}$ cm$^3$ molec$^{-1}$ s$^{-1}$ for the products of
VOC oxidation and P-S/IVOC oxidation respectively (Hayes et al., 2015). For each
oxidation step during aging, there is a mass increase of 7.5 % due to added oxygen.

The gas-particle partitioning is calculated in each bin by using the reformulation
of Pankow theory by Donahue et al. (2006).

$$x_{p,i} = \left(1 + \frac{C_i}{C_{OA}}\right)^{-1} ; C_{OA} = \sum_i [\text{SVOC}]_i \, x_{p,i}$$

Where $\chi_{p,i}$ is the particle phase fraction of lumped species $i$ (expressed as a mass
fraction); $C_i$ is the effective saturation concentration, and $C_{OA}$ is the total mass of organic
aerosol available for partitioning (in μg m$^{-3}$). Only species in the gas phase are allowed to
react with OH radicals in the model, since aerosol species react at much lower rates
(Donahue et al., 2013).

The simulated SOA mass from the model is compared against field measurements
of aerosol composition including results from PMF analysis of aerosol mass spectrometry
data (Hayes et al., 2013; 2015). Specifically, the model predictions of urban SOA (i.e.
SOA formed within the South Coast Air Basin) are compared against the semi-volatile
oxygenated organic aerosol (SV-OOA) concentration from the PMF analysis. The other
OA component also attributed to SOA, low-volatility oxygenated organic aerosol (LV-
OOA), is primarily from precursors emitted outside the South Coast Air Basin and is
used to estimate the background secondary organic aerosol (BG-SOA) as discussed
previously (Hayes et al., 2015).

## 2.4 Correction for changes in partitioning due to emissions into a shallower boundary layer upwind of Pasadena

As described in Hayes et al. (2015), during the transport of the pollutants to Pasadena, the planetary boundary layer (PBL) heights increase during the day. Using CO as a conservative tracer of emissions does not account for how the shallow boundary layer over Los Angeles in the morning influences gas-particle partitioning due to lower vertical mixing and higher absolute POA and SOA concentrations at that time. Thus, as shown in the gas-particle partitioning equation above, there will be a higher partitioning of the species to the particle phase and less gas-phase oxidation of primary and secondary SVOCs. Later in the morning and into the afternoon the PBL height increases (Hayes et al., 2013) diluting the POA and urban SOA mass as photochemical ages increases. However this is a relatively small effect as the partitioning calculation in the SOA model is relatively insensitive to this effect and the absolute OA concentrations (Dzepina et al., 2009; Hayes et al., 2015). Our previous work (Hayes et al., 2015) found in a sensitivity study a +4/-12% variation in predicted urban SOA when various limiting cases were explored for simulation of the PBL (e.g. immediate dilution to the maximum PBL height measured in Pasadena versus a gradual increase during the morning).

To account for the effect of absolute OA mass on the partitioning calculation, the absolute partitioning mass is corrected using the following method. A PBL height of 345 m is used for a photochemical age of 0 h and it reaches a height 855 m at a photochemical age of 9.2 h, which is the maximum age for the ambient field data. Between the two points, the PBL is assumed to increase linearly. The boundary layer heights are determined using ceilometer measurements from Pasadena at 6:00 - 9:00 and 12:00 - 15:00 local time, respectively (Hayes et al., 2013). The second period is chosen because it corresponds to when the maximum photochemical age is observed at the site. The first period is chosen based on transport times calculated for the plume from downtown Los Angeles (Washenfelder et al., 2011) that arrives in Pasadena during the afternoon. There are certain limitations to this correction for the partitioning calculation. First, the correction is based on a conceptual framework in which a plume is emitted and then transported to Pasadena without further addition of POA or SOA precursors. A second limitation is that we do not account for further dilution that may occur as the plume is advected downwind of Pasadena. However, such dilution is not pertinent to the OFR measurements, and so for photochemical ages beyond ambient levels observed at Pasadena, we focus our analysis on the comparison with the OFR measurements.

## 3. RESULTS AND DISCUSSION

## 3.1 Evolution of SOA concentration over 3 days

We follow an approach similar to Hayes et al. (2015) in order to analyse the model results. The model SOA concentration is normalized to the background subtracted CO concentration to account for dilution, and the ratio is then plotted against photochemical age rather than time to remove variations due to diurnal cycles of precursor and oxidant concentrations. The photochemical age is calculated at a reference OH radical concentration of $1.5 \times 10^6$ molec cm$^{-3}$ (DeCarlo et al., 2010). Figure 3 shows this analysis for each model case for up to 3 days of photochemical aging. Since fragmentation and dry deposition are not included in the model, it has only been run to 3 days in order to minimize the importance of these processes with respect to SOA concentrations (Ortega et al., 2016). Nevertheless, it is very likely that gas-phase fragmentation of SVOCs (e.g. branching between functionalization and fragmentation) occurs during oxidative aging over these photochemical ages as is discussed in further detail below.

In each panel of Fig. 3, field measurements are included for comparison. The ambient urban SOA mass at the Pasadena ground site is generally measured under conditions corresponding to photochemical ages of 0.5 days or less (Hayes et al., 2013). The airborne observations of SOA in the Los Angeles basin outflow are also shown as the average of all data between 1 and 2 days of photochemical aging (Bahreini et al., 2012). The gray region on the right serves as an estimate for very aged urban SOA based on data reported by de Gouw and Jimenez (2009). The data from the OFR and a fit of the ambient and reactor data (dotted black line) are also displayed in Fig. 3 (Ortega et al., 2016). In addition, Figure 4 shows the ratio of modeled-to-measured SOA mass on a logarithmic axis to facilitate evaluation of model performance.

As displayed in the graphs for Fig. 3, it should be noted the measurements from the OFR (Ortega et al., 2016) and from the NOAA P3 research aircraft (Bahreini et al., 2012) give quite similar results for SOA/ΔCO. The OFR measurements are not affected by particle deposition that would occur in the atmosphere at long timescales or photochemical ages. Only a few percent of the particles are lost to the walls of the reactor, and this process has been corrected for already in the results of Ortega et al. The similarity in the two types of observations suggests that ambient particle deposition and plume dispersion do not significantly change the SOA/ΔCO ratio over the photochemical ages analyzed here.

In ROB + TSI, as described in previous work (Hayes et al., 2015), there is a large over-prediction of SOA mass at longer photochemical ages. As displayed in Fig. 3, the amount of SOA produced in the model is higher than all of the field measurements taken

at a photochemical age longer than 0.5 days. Moreover, the ratios of model to measurement are higher than the upper limit of the gray bar representing the ratios within the measurement uncertainties. There is an agreement with the measurements at moderate photochemical ages (between 0.25 and 0.50 days), but the SOA mass simulated by the model is slightly lower than the measurements at the shortest photochemical ages (less than 0.25 days) even when accounting for measurement uncertainties. In this parameterization, most of the SOA produced comes from the P-S/IVOCs, and uncertainties in the model with respect to these compounds likely explain the overestimation observed at longer photochemical ages. As discussed in the introduction, a major goal in this work is to better constrain the amount of SOA formed from the oxidation of P-S/IVOCs, and the following two model cases (ROB + ZHAO + TSI and WOR + ZHAO + TSI) seek to incorporate new measurements to better constrain the box model with respect to the P-S/IVOCs.

When the yield, rate constants, and initial concentrations of P-IVOCs are constrained using the field measurements reported in Zhao et al. (2014) (ROB + ZHAO + TSI), the SOA mass simulated by the model shows much better agreement with the measurements at longer photochemical ages (Fig. 3 and 4). There is a slight over-prediction at 2 days of photochemical aging, but the model is still within the range of measurements of very aged urban SOA reported by De Gouw and Jimenez (2009). The parameterization reported in Robinson et al. (2007) for P-S/IVOCs is based on one study of the photo-oxidation of diesel emissions from a generator (Robinson et al., 2007). The results obtained here for the better constrained ROB + ZHAO + TSI case indicate that the initial concentrations of P-IVOCs as well as the P-IVOC yields within ROB + TSI are too high which leads to over-prediction of SOA concentration at longer photochemical ages. On the other hand, the SOA mass simulated in ROB + ZHAO + TSI is biased low at shorter photochemical ages (less than 1 day). Similar to other recent studies (Gentner et al., 2012; Hayes et al., 2015; Ortega et al., 2016), there may be unexplained SOA precursors not included in the model which rapidly form SOA or yields for fast-reacting species including certain VOCs may be biased low. Both of these possibilities are explored in the other model cases discussed below.

The WOR + ZHAO + TSI case simulates higher SOA concentrations at shorter photochemical ages compared to the previous case (ROB + ZHAO + TSI), but it is still biased low at shorter photochemical ages. The more rapid SOA formation is due to the updated SVOC volatility distribution in this model case compared to the cases that use the Robinson et al. (2007) distribution. Specifically, as shown in Fig. 2F, there is a higher relative concentration of gas phase SVOCs in the $c^* = 10^2$ bin and thus a higher ratio of P-SVOC to POA. Given that in the box model (and in most air quality models) the P-SVOC emissions are determined by scaling the POA emissions according to their volatility distribution, a higher P-SVOC to POA ratio will then result in a higher initial P-

SVOCs concentration. Furthermore, SOA formation from P-SVOCs is relatively fast.
Together these changes lead to increases in SOA formation during the first hours of
photochemical aging when using the Worton et al. volatility distribution. This case
suggests that P-SVOCs in their highest volatility bin ($c* = 10^2$ µg m$^{-3}$ bin) that are
emitted by motor vehicles may be responsible for some of the observed rapid SOA
formation within the South Coast Air Basin. When observing the SOA mass simulated at
photochemical ages higher than 1 day, the simulation is similar to ROB + ZHAO + TSI.
There is better model/measurement agreement than for the ROB + TSI case, but a small
over-prediction is observed in the comparison to the reactor data at 2 days of
photochemical aging.
Also shown in the right-hand panels of Fig. 3 and 4 are the results with the
updated yields for the VOCs that account for gas phase chamber wall losses. For these
last three cases (ROB + MA, ROB + ZHAO + MA, and WOR + ZHAO + MA), the rate
of SOA formation at short photochemical ages is faster because the secondary SVOC
mass from the oxidation of the VOC precursors is distributed into lower volatility bins
compared to the Tsimpidi et al. (2010) parameterization. In the ROB + MA case (Fig. 3D
and 4D), similar to ROB + TSI, an over-prediction is obtained at longer photochemical
ages. There is an improvement in the model at the shortest photochemical ages, but the
simulated mass is still lower than the measurements even when considering the
measurement uncertainty. Both of these cases perform less well for SOA formation
within the South Coast Air Basin, and therefore the remainder of this study is focused on
the other four model cases. Overall, the model cases using the updated yields for V-SOA
show improvement for the shorter photochemical ages, and the evolution of SOA
concentration as a function of photochemical age better corresponds to the various
measurements taken at Pasadena and from the OFR.
Specifically, the ROB + ZHAO + MA and the WOR + ZHAO + MA cases both
better represent SOA formation and exhibit better model/measurement agreement among
the different cases used in this work. They are both consistent with the OFR reactor data
at longer photochemical ages as shown in Figs. 3 and 4 compared with the other cases. At
a qualitative level, the MA parameterization simulations are more consistent with the fit
of the OFR measurements in which the SOA mass remains nearly constant at longer
photochemical ages. In contrast, the cases with the TSI parameterization do not follow
this trend as the SOA mass keeps increasing between 2 and 3 days age, which is not
observed in the measurements. As already mentioned, the model used for this work does
not include fragmentation reactions, and including these reactions, in particular branching
between functionalization and fragmentation during gas-phase SVOC oxidation, may
improve the cases using a potential update of the TSI parameterization as discussed
below. Fig. 4F indicates that including additional P-SVOC mass in the model and
accounting for gas-phase wall losses in chamber studies improves SOA mass
concentration simulations with respect to the measurements. However, in the WOR +
ZHAO + MA case there is still a slight under-prediction of SOA formed at shorter
photochemical ages (between 0.05 and 0.5 days), and this discrepancy is observed in all
the other model cases. Given the uncertainties in the model set-up discussed in the
experimental section, it is not possible to conclude if one of the four cases (i.e. ROB +
ZHAO + TSI, WOR + ZHAO + TSI, ROB + ZHAO + MA, WOR + ZHAO + MA) more
accurately represents SOA formation in the atmosphere.
According to the OFR data from Ortega et al. (2016), the mass of OA starts to
decay due to fragmentation after heterogeneous oxidation at approximately 10 days of
photochemical aging. The results are consistent with other OFR field measurements
(George and Abbatt, 2010; Hu et al., 2016; Palm et al., 2016). In this work, the model is
run only up to 3 days, which is much shorter than the age when heterogeneous oxidation
appears to become important. In fact, when including a fragmentation pathway for each
of the model cases, a reduction of OA of only 6 % is observed compared to the cases
without fragmentation at 3 days of photochemical aging. In this sensitivity study, the
fragmentation is parameterized as an exponential decrease in OA concentration that has a
lifetime of 50 days following Ortega et al. (2016). Given the results, the inclusion of
fragmentation due to heterogeneous oxidation in the model does not significantly change
the model results or the conclusions made in this work.
More generally, there are at least three different fragmentation mechanisms that
could be responsible for the decrease of SOA formation at very high photochemical ages.
The first mechanism is the reaction of oxidants (e.g. OH) with the surface of an aerosol
particle and decomposition to form products with higher volatility, i.e., due to the
heterogeneous oxidation just described. The second type of fragmentation that may be
important for very high photochemical ages in the OFR is due to the high concentration
of OH (Palm et al., 2016). Most of the molecules in the gas phase will react multiple
times with the available oxidants before having a chance to condense, which will lead to
the formation of smaller products too volatile to form SOA. However, this is only
important at very high photochemical ages in the OFR, which are not used in this work.
A third type of fragmentation can occur during the aging of gas-phase SVOCs
(Shrivastava et al., 2013; 2015). The TSI parameterization used in the model from this
work and from previous modeling works (Robinson et al., 2007; Hodzic et al., 2010;
Shrivastava et al., 2011) only includes the functionalization of the SVOCs and neglects
fragmentation reactions. More recently, Shrivastava et al. (2013) have modified the VBS
approach in a box model by incorporating both pathways and performed several
sensitivity studies. The results when including fragmentation generally exhibit better
agreement with field observations, but as noted in that work the agreement may be
fortuitous given that both the emissions as well as the parameters representing oxidation
in the model are uncertain. This third type of fragmentation is not simulated in our
sensitivity study using the approach above, and it remains poorly characterized due to the
complexity of the chemical pathways and the number of compounds contributing to SOA
formation as described in Shrivastava et al. (2013).

Despite having higher SOA yields initially, over regional scales (i.e.
photochemical ages at and above approximately 2 days) the parameterizations with
updated V-SOA yields and without aging produce less SOA, because the organic mass in
higher volatility bins ($c^* = 100$ and $1000$ $\mu g$ $m^{-3}$) is not further oxidized by aging
reactions to produce organics with sufficiently low volatilities to form SOA (Fig. S1 –
S7). Furthermore, large SOA overpredictions have been shown to occur in gridded 3-D
models when using parameterizations with aging that do not include fragmentation
reactions (Shrivastava et al., 2015). Fragmentation with aging reactions may still play a
role in determining SOA concentrations on such regional scales. However for the
photochemical ages studied here, our results as well as the recent findings regarding gas-
phase wall losses in chamber studies, suggest the inclusion of updated V-SOA yields as
well as accurate parameterizations for I-SOA and S-SOA and for the emissions of
precursors is more important for accurately predicting urban SOA concentrations.

Finally, Woody et al. (2016) recently proposed a meat cooking volatility
distribution and therefore we perform a sensitivity study by using this distribution in our
model for P-SVOCs coming from cooking sources. The results are displayed in the
supporting information (Figure S8), where this alternate approach has been implemented
for the WOR + ZHAO + TSI and WOR + ZHAO + MA cases. By comparing the results
obtained from this sensitivity study with Fig. 3, the two cases in the sensitivity study
display a slight decrease of SOA/$\Delta$CO values over 3 days of photochemical aging with a
difference of approximately 9 % at 3 days. Thus, the model-measurement comparison
does not change significantly relative to the base case. Given the similarities between the
sensitivity study and Fig. 3, as well as the possibility of cooking SOA sources other than
meat-cooking (i.e. heated cooking oils, Liu et al. (2017)), the remainder of our work uses
the Robinson et al. volatility distribution for P-SVOCs from cooking sources.

### 3.1.1 SOA concentration estimated at Pasadena: fossil and non-fossil fractions

In the top panel of Figure 5, the box model is compared against the urban SOA
determined by PMF analysis of the AMS measurements at Pasadena (Hayes et al., 2013).
In the bottom panel of the same figure the model is compared against the fossil and non-
fossil fraction of urban SOA as obtained from [14]C measurements reported in Zotter et al.
(2014). Both panels show measurements and predictions corresponding to 12:00 – 15:00
local time, when SOA concentrations peaked due to longer photochemical ages (5 hours
on average) as well as the arrival of emissions transported from source-rich western
regions of the South Coast Air Basin.

Similar to the results in Fig. 3 and 4 for short photochemical ages, the SOA mass simulated by the ROB + ZHAO + TSI case is biased low in Fig. 5A. The ROB + ZHAO + MA, WOR + ZHAO + TSI, and WOR + ZHAO + MA cases show better model/measurement agreement as the simulated SOA mass is within the measurement uncertainty or essentially equal to the lower limit of the concentration that is defined by the measurement uncertainty. Fig. 5A also allows evaluation of the contribution of each precursor type to the SOA at Pasadena. For the four cases displayed, the P-SVOCs and P-IVOCs are responsible for 70 – 83 % of the urban SOA formation. Thus, more than half of the urban SOA is attributed to these precursors even in the MA parameterizations where the model is run with the updated yields, which doubles V-SOA compared to the cases using the yields reported from Tsimpidi et al. (2010). Furthermore, 8 – 27 % of the measured urban SOA is due to V-SOA where the range of values is due to the uncertainty in the measurements as well as the difference in simulated V-SOA concentration for each case.

According to the [14]C measurements, an average of 71 ± 3 % of urban SOA at Pasadena is fossil carbon, which is thought to be due to the importance of vehicular emissions, especially during the morning rush hour (Bahreini et al., 2012; Zotter et al., 2014; Hayes et al., 2015). In general, the box model gives results consistent with the [14]C measurements. To make this comparison, the simulated SOA is apportioned between fossil S-SOA, fossil I-SOA, fossil V-SOA, cooking S-SOA, and biogenic V-SOA. The last two apportionments correspond to non-fossil carbon. This evaluation is possible following an approach similar to Hayes et al. (2015) where the identity of the precursor is used to apportion SOA. Briefly, the fossil S-SOA is formed from P-SVOCs emitted with hydrocarbon-like OA (HOA), which is a surrogate for vehicular POA. Second, cooking S-SOA is formed from P-SVOCs emitted with cooking-influenced OA (CIOA). The concentrations of HOA and CIOA were determined previously using PMF analysis. Fossil V-SOA is formed from aromatics, alkanes, and olefins while isoprene and terpenes are responsible for biogenic V-SOA. The treatment of IVOCs in the comparison with the [14]C measurements has been updated from our 2015 study. Previously, it was assumed that P-IVOCs were co-emitted with cooking-influenced OA, but the recent work of Zhao et al. (2014) and others indicates that petroleum sources contribute substantially to IVOC emissions (Dunmore et al., 2015; Ots et al., 2016). Therefore, the IVOCs are considered entirely fossil carbon in order to obtain the results shown in Fig. 5B.

As seen in Fig. 5B, for all the model cases, cooking S-SOA dominates the non-fossil fraction and biogenic VOCs have only a small contribution to non-fossil urban SOA. This result is consistent with our previous work, and indicates agreement between the model and [14]C measurements cannot be achieved without including an urban source of non-fossil carbon such as P-SVOCs from cooking. With respect to fossil SOA, more S-SOA is formed when using the volatility distribution of vehicular POA reported from

Worton et al. (2014) due to the greater proportion of gas-phase of P-SVOCs. When the
V-SOA yields are updated in the model (MA parameterizations), there is a corresponding
increase in both fossil and non-fossil V-SOA.
When comparing the fossil/non-fossil carbon split, all the cases are either in
agreement with the measurement within its uncertainty, or slightly lower. Starting with
the ROB + ZHAO + TSI case, the fossil fraction increases from 75 % to 79 % in each
case as VOCs or P-SVOCs from vehicle emissions have greater importance for SOA
formation. While the uncertainties reported in Zotter et al. (2014) were 71 ± 3 %, there
are likely additional errors due to different factors that may influence the model or
measurements. For example, a portion of the P-IVOCs may be from cooking sources
rather than entirely from fossil sources as is assumed above (Klein et al., 2016). Taking
the WOR + ZHAO + MA case as an example, since it is the best performing case in this
work according to Fig. 5A, model/measurement agreement is obtained within
measurement uncertainties if one assumes that 19 – 39 % of P-IVOCs come from
cooking emissions. Ultimately, the differences observed in the comparison with the $^{14}$C
data are very likely smaller than these errors discussed here, and it is concluded that all
the model cases perform equally well with respect to the fossil/non-fossil carbon split.
As reported in Gentner et al. (2012), emissions from petroleum derived fuels such
as diesel and gasoline have an important contribution to the formation of SOA. However,
there have been conflicting results regarding the relative contributions of diesel versus
gasoline emissions (Bahreini et al., 2012; Gentner et al., 2012). In this work, the relative
contribution of different SOA sources is estimated following a procedure similar to that
previously published in Hayes et al. (2015), and the results are shown in Figure S9 of the
supporting information. Briefly, the source apportionment method follows four steps.
First, after classifying the SOA mass from isoprene and terpenes as biogenic V-SOA, the
remaining V-SOA is attributed to gasoline emissions since the diesel contribution to V-
SOA is small (~3 %) (Hayes et al., 2015). Second, for the diesel and gasoline
contribution to S-SOA, 70(±10) % of HOA is emitted from diesel vehicles with the
remainder from gasoline vehicles (Hayes et al., 2013), and thus it is assumed for the
source apportionment that 70% (30%) of vehicular P-SVOCs are from diesel (gasoline)
vehicles. Third, the S-SOA from cooking sources is calculated separately in the model,
where the initial concentration of cooking P-SVOCs is estimated using the measured
CIOA concentration and the method described in Section 2.2.2 above. Lastly, the
fractional contributions to I-SOA mass is difficult to determine since there are still
uncertainties about the sources of IVOCs. According to Zhao et al. (2014), petroleum
sources other than on-road vehicles likely contribute substantially to primary IVOCs, but
evidence exists that cooking may be a source of IVOCs as well (Klein et al., 2016). Thus,
while we attribute I-SOA to these two sources, we do not distinguish the sources. The
estimated source apportionment in Fig. S9 attributes urban SOA as follows: 4% to

biogenic V-SOA, 22% to gasoline V-SOA, 9% to gasoline S-SOA, 20 % to diesel S-SOA, and 16 % to cooking S-SOA. The remaining 29 % is I-SOA that is either due to cooking or off-road emissions of P-IVOCs.

It should be noted that according to McDonald et al. (2015), the emissions from vehicles have decreased over time (i.e. between 1970 and 2010) due to regulations in California. Warneke et al. (2012) have observed also that the emission ratios of some SOA precursors (i.e. $\Delta VOC/\Delta CO$) have remained constant between 2002 and 2010, while absolute concentrations have decreased. On the other hand, cooking and off-road emissions are subject to different regulations in California, and the ratios of cooking or off-road emissions to vehicular emissions have likely changed with time, which means that the source apportionment results for urban SOA presented here will be specific to 2010.

## 3.2 SOA formation versus precursor oxidation rate constant

Recent results from Ortega et al. (2016) point to the importance of fast-reacting precursors for urban SOA during CalNex, and we can use their results to further evaluate our box model. The fraction of SOA formed from each precursor class as a function of the precursor rate constant is displayed in Figure 6. The right-axis of Fig. 6 shows the correlation ($R^2$) of different VOCs with the maximum concentration of SOA formed using the OFR as a function of their oxidation rate constants as reported in Ortega et al. (2016). This analysis of the OFR data allows us to constrain the rate constants of the most important SOA precursors. A detailed description of how the $R^2$ values were obtained can be found in Ortega et al. (2016). According to the $R^2$ data, the VOC compounds that correlate best with maximum SOA formation potential are those that have log $k_{OH}$ rate constants ranging from -10.5 to -10.0. When comparing the percentage of SOA mass simulated by the model with the observed $R^2$ values, all of the four cases are not entirely consistent with the $R^2$ data. According to the model, more SOA mass is formed from precursors in the bin ranging from -11.0 to -10.5 (the majority of mass formed comes from P-IVOCs) rather than the bin ranging from -10.5 to -10.0. In contrast, the $R^2$ value is higher for the more reactive bin. If either fast-reacting precursors were missing in the model, or if the rate constants of the currently-implemented precursors were too small, then correcting either error would shift the relative distribution shown in Fig. 6 towards faster-reacting SOA precursors. In turn, the trend in the percentage of modeled SOA mass may more closely follow the trend in $R^2$ values.

## 3.3 Volatility distribution of OA

Based on the evaluations carried out up to this point on the six model cases, the WOR + ZHAO + MA case seems to most closely reproduce the observations. Thus, the entire volatility distribution of the OA, precursors, and secondary gas phase organics is analyzed for this model case. Figure 7 shows this distribution for three selected photochemical ages: 0, 5, and 36 h. The figure allows us to track the evolution of SOA and secondary gas phase organics from each precursor class in terms of their concentration and volatility and also to evaluate the reduction of precursor concentrations. For the model results, the volatility distribution of all organics resolved by precursor class, except for the VOCs and P-IVOCs, can be taken directly from the model. To determine the volatility distribution of the VOCs and P-IVOCs, the SIMPOL.1 method (Pankow and Asher, 2008) is used to estimate the effective saturation concentration of each compound or lumped species in the model. Also included in Fig. 7, in the bottom-right panel, is the observed volatility distribution for the Pasadena ground site, which is an average of measurements collected during 12:00 – 15:00 local time and corresponds to 5 h of photochemical aging. For the measurements, the volatility distribution of VOCs was determined using GC-MS data (Borbon et al., 2013) whereas the IVOC distribution is taken from Zhao et al. (2014). The volatility distribution of SVOCs was determined using combined thermal denuder AMS measurements (see the supporting information for further details).

For the volatility distribution of the model at time 0, the concentrations of P-SVOCs and P-IVOCs monotonically increases with the value of $c^*$. However, a discontinuity in the mass concentration exists between the $c^* = 10^2$ and $10^3$ µg m$^{-3}$ bins. This discontinuity can be explained by several factors. First, the measured IVOCs mass concentration the $c^* = 10^3$ µg m$^{-3}$ bin is very low, and since the initial concentrations of IVOCs in the model are constrained by the field measurements, the model will also have very low concentrations. Zhao et al. (2014) have already noted that the concentration of P-IVOCs in this bin is relatively low when compared to the volatility distribution from Robinson et al. (2007). Another possible explanation is the presence of cooking sources, which in the model are responsible for substantial P-SVOC mass (~50%) but may have a smaller contribution to the P-IVOC mass.

During oxidation the volatility distribution evolves and the concentration of secondary organics increases in the bins between $c^* = 10^{-1}$ and $10^3$ µg m$^{-3}$ (inclusive), and the largest portion of SOA is found in the $c^* = 1$ µg m$^{-3}$ bin. This result is due to the partitioning of the organic mass to the particle phase and the lack of particle phase reactions in the model, which leads to very slow oxidation rates for species found in the lower volatility bins. After 36 h, a large portion of the precursors have been reacted,

although some primary and secondary material remains in the gas phase giving rise to
more gradual SOA formation.

In Fig. 7, it is possible to compare the measured volatility distribution with the
model simulation at 5 h of photochemical aging. It should be noted that the relatively
high concentrations of VOCs in the model compared to the measurements are due to the
model containing VOCs for which measurements were not obtained in Pasadena. There
are 47 VOCs used in the model and only 19 VOCs were measured. However, the
remaining VOCs have been measured in other urban locations (Warneke et al., 2007;
Borbon et al., 2013) and thus it is assumed they are also present in the South Coast Air
Basin. For this work, we include these 28 remaining VOCs by assuming that they are also
emitted in the South Coast Air Basin with identical emission ratios ($\Delta$VOC/$\Delta$CO). When
comparing only VOCs measured and modeled (shown in hollow black bars), the results
are consistent (3.1, 3.6 and 2.2 $\mu$g m$^{-3}$ from c* = $10^7$ to $10^9$ $\mu$g m$^{-3}$ bins versus 3.8, 3.7
and 2.2 $\mu$g m$^{-3}$ for the measurements). On the other hand, the model appears to have a
low bias for the concentrations of P-IVOCs (0.16, 0.63, 0.89 and 2.3 $\mu$g m$^{-3}$ $^3$ from c* =
$10^3$ to $10^6$ $\mu$g m$^{-3}$ bins versus 0.21, 1.39, 2.65 and 3.82 $\mu$g m$^{-3}$ for the measurements).
This low bias is seen for each volatility bin and could possibly be explained by either
oxidation rate constants that are too high or $\Delta$IVOC/$\Delta$CO ratios that are too low. The
latter explanation seems more likely given that the rate constants estimated using
surrogate compounds and structure-activity relationships for the unspeciated P-IVOCs
are generally lower limits (Zhao et al., 2014), which would result in a high bias rather
than a low bias. The $\Delta$IVOC/$\Delta$CO ratios may be low because the photochemical age
between 00:00 – 6:00 local time is not strictly zero, and some oxidation may have
occurred during the period used to determine the ratio values. Emission ratios such as
$\Delta$IVOC/$\Delta$CO facilitate incorporating P-IVOC emissions into 3-D models that already use
CO emissions inventories, and the $\Delta$IVOC/$\Delta$CO ratios reported here could be used for
this purpose. However, the resulting I-SOA concentrations should be considered lower
limits given that the emission ratios, and also the rate constants, are likely themselves
lower limits.

To further explore the impact of potential errors in the initial IVOC
concentrations, a sensitivity study has been carried out using initial concentrations
calculated based on the observed photochemical age and measured IVOC concentrations
at Pasadena as well as the estimated IVOC oxidation rate constants (Zhao et al., 2014).
This alternate approach is implemented for the ROB + ZHAO + MA and WOR + ZHAO
+ MA cases and does not use nighttime IVOC-to-CO ratios. The results when using this
alternative approach are shown in the supporting information (Figure S10). When
comparing Fig. S10 with Fig. 3, differences are minor. The model/measurement
agreement improves slightly at shorter photochemical ages (less than 1 day). At the same
time a slightly larger over-prediction is observed at longer photochemical ages. However,
the formation of SOA modeled in this sensitivity test is similar to the original cases from
Fig. 3 with an average difference of only 21 %, which represent a relatively small error
compared to other uncertainties in SOA modeling. The IVOC initial concentrations used
in this sensitivity test are slightly higher than those calculated using the IVOC-to-CO
ratio, which explain the small increase of modeled SOA/$\Delta$CO. Ultimately, the different
approaches for determining the initial IVOC concentration in the model are reasonably
consistent, and both approaches perform similarly given the model and measurement
uncertainties.

For the measurements of SVOCs, all the mass in bins lower than $10^{-2}$ $\mu$g m$^{-3}$ are

lumped into this bin for Fig. 7 since the model does not contain lower volatility bins. In
addition, the $10^1$ and $10^2$ $\mu$g m$^{-3}$ bins are not well-resolved because the thermal denuder
did not consistently reach temperatures low enough (less than 37°C) to resolve SVOCs in
this range of volatilities. Thus, the $10^1$ $\mu$g m$^{-3}$ bin may contain some higher volatility
particulate mass although this contribution is expected to be small due to the low particle
phase fraction of compounds in the $10^2$ $\mu$g m$^{-3}$ bin. With these considerations in mind, the
volatility distribution of the SVOCs is somewhat different in the model compared to the
measurements. Most notably, the model does not form a significant amount of lower
volatility SOA in the $10^{-2}$ $\mu$g m$^{-3}$ bin, whereas the measurements have a much higher
concentrations in this bin. A factor that may explain this difference between the volatility
distributions is the lack of particle phase reactions that continue to transform SOA into
lower volatility products, a process which is not considered in the model. One example of
a particle phase reaction is the formation of SOA within deliquesced particles, including
the partitioning of glyoxal to the aqueous phase to produce oligomers as discussed in
Ervens and Volkamer (2010), although that specific mechanism was of little significance
during CalNex (Washenfelder et al., 2011; Knote et al., 2014). Alternatively, the use of
an aging parameterization where the volatility may decrease by more than one order of
magnitude per oxidation reaction would also distribute some SOA mass into lower c*
bins. Hayes et al. (2015) previously evaluated different parameters for aging. However,
the results from this previous study showed that substantial over-prediction of SOA was
observed when using the Grieshop et al. (2009) parameterization in which each oxidation
reaction reduced volatility by two orders of magnitude. New parameterizations may be
necessary to produce the observed SOA volatility and concentration simultaneously
(Cappa and Wilson, 2012). However, we note that the additional low volatility organic
mass will not significantly change SOA predictions in urban regions where OA
concentrations are relatively high. When comparing the total amount of particle phase
SVOCs, it seems that the model reproduces reasonably well the measurements (6.2
versus 9.0 $\mu$g m$^{-3}$) as expected based on the comparisons of the total SOA concentration
discussed above. In addition, the total amount of SVOCs (particle and gas phase) are
similar (11.2 vs 11.8 $\mu$g m$^{-3}$), although it is difficult to determine from measurements the
gas phase concentration of SVOCs in the $10^2$ μg m$^{-3}$ bin due to the lack of particle mass
in this bin under ambient concentrations as well as the limited temperature range of the
thermal denuder system.
Recently, Woody et al. (2016) published a paper that modeled SOA over
California using the Environmental Protection Agency's Community Multiscale Air
Quality Model that had been updated to include a VBS treatment of SOA (CMAQ-VBS).
As discussed in that paper, the modeled P-S/IVOC emission inventories remain an
important source of uncertainty in 3-D grid-based models. In that previous study several
different ratios of P-S/IVOCs-to-POA emissions were evaluated against measurements,
and it was found that a ratio of 7.5 gave the best agreement between the CMAQ-VBS
model and observations. From the results shown in Fig. 7 at a photochemical age of 0 h, a
P-S/IVOC-to-POA ratio of 5.2 is calculated. This ratio is different from that determined
by Woody et al. (2016), and may be biased low due to possibly low ΔIVOC/ΔCO
emission ratios as discussed earlier in this section, but it serves as both a useful lower
bound and has the advantage of being determined from empirical measurements of
aerosols rather than by tuning a model to match measured SOA concentrations. As stated
in Woody et al. (2016), the higher ratio may compensate for other missing (or
underrepresented) formation pathways in SOA models or excessive dispersion of SOA in
their model.
**4. CONCLUSION**
We have used several data sets from recently published papers to better constrain
and evaluate urban SOA formation pathways and precursors, especially P-SVOCs and P-
IVOCs, within a custom-built box model. The use of the box model facilitates the
incorporation of these new data sets as well as the evaluation of a number of model cases.
All the model cases are able to correctly simulate the fossil/non-fossil carbon split at the
Pasadena ground site providing support for the performance of the model. When
measurements of IVOCs are used to constrain the concentrations of P-IVOCs, such as in
the ROB + ZHAO + TSI and ROB + ZHAO + MA cases, a large improvement of the
model at longer photochemical age is observed. However, these model cases are still
biased low at shorter photochemical ages. By constraining the P-SVOCs additionally
with measurements of those precursors, such as in the WOR + ZHAO + TSI case, better
model/measurement agreement is obtained at shorter photochemical ages, yet the model
is still biased low. Finally, the WOR + ZHAO + MA case, which incorporates state-of-
the-art measurements of P-SVOCs and P-IVOCs and also accounts for the effect of
chamber wall-losses on VOC yields, obtains model/measurement agreement within
measurement uncertainties at long photochemical ages. Although, it displays also a low
bias at short photochemical ages, which is similar to the ROB + ZHAO + MA case. This
bias may be due to low ΔIVOC/ΔCO emissions ratios or IVOC oxidation rate constants
for which the estimated values are too low. It is also possible that additional sources or
SOA formation pathways are missing from the model. Moreover, a P-S/IVOC-to-POA
ratio of 5.2 is determined, which can be combined with POA emission inventories to
constrain the emissions of P-S/IVOCs in gridded chemical transport models.

In addition to evaluating the model performance with respect to SOA
concentration, the rates of SOA formation are compared against measurements as well.
This aspect of the study was enhanced by the use of OFR data to constrain SOA
formation potential for up to 3 days of photochemical aging (at $1.5 \times 10^6$ molec OH
$cm^{-3}$). The model cases that include multi-generation oxidative aging predict substantial
SOA increases after 1.5 days of aging, which is not consistent with the OFR
measurements. In contrast, model cases in which aging is omitted and instead SOA yields
for VOCs are corrected for gas phase wall-losses in chamber experiments predict little
change in the SOA concentration after 1.5 days. These results highlight the uncertainties
associated with aging schemes for SOA from VOCs, which are often implemented in
SOA models. Implementing instead corrected yields for VOCs results in similar amounts
of SOA but formation rates versus time that are more consistent with observations.

Therefore, the model cases with updated VOC yields that account for chamber
wall-losses best reproduce the ambient and OFR data. However, while the WOR +
ZHAO + MA case appears to represent a slight improvement over the ROB + ZHAO +
MA case, as well as over the ROB + ZHAO + TSI and WOR + ZHAO + TSI cases, it is
not possible to conclude that one set of parameters is better than the other since the
difference in the predictions for these 4 cases (15 % on average) is likely smaller than the
uncertainties due to the model setup as well as the lack of a gas-phase fragmentation
pathway during aging. Moreover, uncertainties in the vapor wall-loss corrected yields
remain, and the correction of the yields has been performed here using data from a
limited number of laboratory studies. In particular, the effect of temperature and humidity
on gas-wall partitioning needs to be characterized. The results obtained in our work
motivate future studies by showing that SOA models using wall-loss corrected yields
reproduce observations for a range of photochemical ages at a level of accuracy that it is
as good as or better than parameterizations with the uncorrected yields.

In all six of the model cases, a large majority of the urban SOA at Pasadena is the
result of P-SVOC and P-IVOC oxidation. While this result alone cannot be taken as
conclusive due to the uncertainties in the model parameters, further evidence for the
importance of P-SVOCs and P-IVOCs is obtained by analyzing the percentage of SOA
formed at long photochemical ages (~1.5 days) as a function of the precursor rate
constant. The P-SVOCs and P-IVOCs have rate constants that are similar to highly
reactive VOCs that have been previously found to strongly correlate with SOA formation
potential measured by the OFR.

Lastly, the modeled volatility distribution of the total (gas and particle phase) organic mass between $c* = 10^{-2}$ and $10^{10}$ ug m$^{-3}$ is analyzed at three ages and compared against volatility-resolved measurements. While the total concentrations of gas and particle phase SVOCs are reasonably well simulated, at the same time there are important differences between the measured and modeled volatility distribution of SVOCs. These differences highlight the need for further studies of the chemical pathways that may give rise to SOA in low volatility bins at $c* = 10^{-2}$ ug m$^{-3}$ and lower.

## ACKNOWLEDGEMENTS

This work was partially supported by a Natural Science and Engineering Research Council of Canada (NSERC) Discovery Grant (RGPIN/05002-2014), le Fonds de recherche - Nature et technologies (FRQNT) du Québec (2016-PR-192364), and the Université de Montréal. AMO and JLJ were supported by CARB 11-305 and EPA STAR 83587701-0. This manuscript has not been reviewed by EPA and thus no endorsement should be inferred. We gratefully acknowledge VOC data provided by J. de Gouw and J.B. Gilman.

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

**Table 1.** Summary of the model cases used in this paper.

| Case | Notes | References |
|---|---|---|
| 1) ROB + TSI | P-S/IVOCs: Robinson et al. parameterization, and all SOA treated within VBS framework<br><br>VOCs: Tsimpidi et al. parameterization with aging | Hayes et al. (2015)<br>Robinson et al. (2007)<br>Tsimpidi et al. (2010) |
| 2) ROB + ZHAO + TSI | P-SVOCs: Robinson et al. parameterization, and all SOA treated within VBS framework<br><br>P-IVOCs: Zhao et al. parameterization with aging<br><br>VOCs: Tsimpidi et al. parameterization with aging | Robinson et al. (2007)<br>Zhao et al. (2014)<br>Tsimpidi et al. (2010) |
| 3) WOR + ZHAO + TSI | P-SVOCs: Worton et al. volatility distribution for vehicular P-SVOCs, Robinson et al. volatility distribution for cooking P-SVOCs<br><br>P-IVOCs: Zhao et al. parameterization with aging<br><br>VOCs: Tsimpidi et al. parameterization with aging | Robinson et al. (2007)<br>Worton et al. (2014)<br>Zhao et al. (2014)<br>Tsimpidi et al. (2010) |
| 4) ROB + MA | P-S/IVOCs: Robinson et al. parameterization, and all SOA treated within VBS framework<br><br>VOCs: VOCs yields corrected for wall-losses, no aging of VOC oxidation products | Robinson et al. (2007)<br>This work |
| 5) ROB + ZHAO + MA | P-SVOCs: Robinson et al. parameterization, and all SOA treated within VBS framework<br><br>P-IVOCs: Zhao et al. IVOC parameterization with aging<br><br>VOCs: VOCs yields corrected for wall-losses, no aging of VOC oxidation products | Robinson et al. (2007)<br>Zhao et al. (2014)<br>This work |
| 6) WOR + ZHAO + MA | P-SVOCs: Worton et al. volatility distribution for vehicular P-SVOCs, Robinson et al. volatility distribution for cooking P-SVOCs<br><br>P-IVOCs: Zhao et al. IVOC parameterization with aging<br><br>VOCs: VOCs yields corrected for wall-losses, no aging of VOC oxidation products | Robinson et al. (2007)<br>Worton et al. (2014)<br>Zhao et al. (2014)<br>This work |


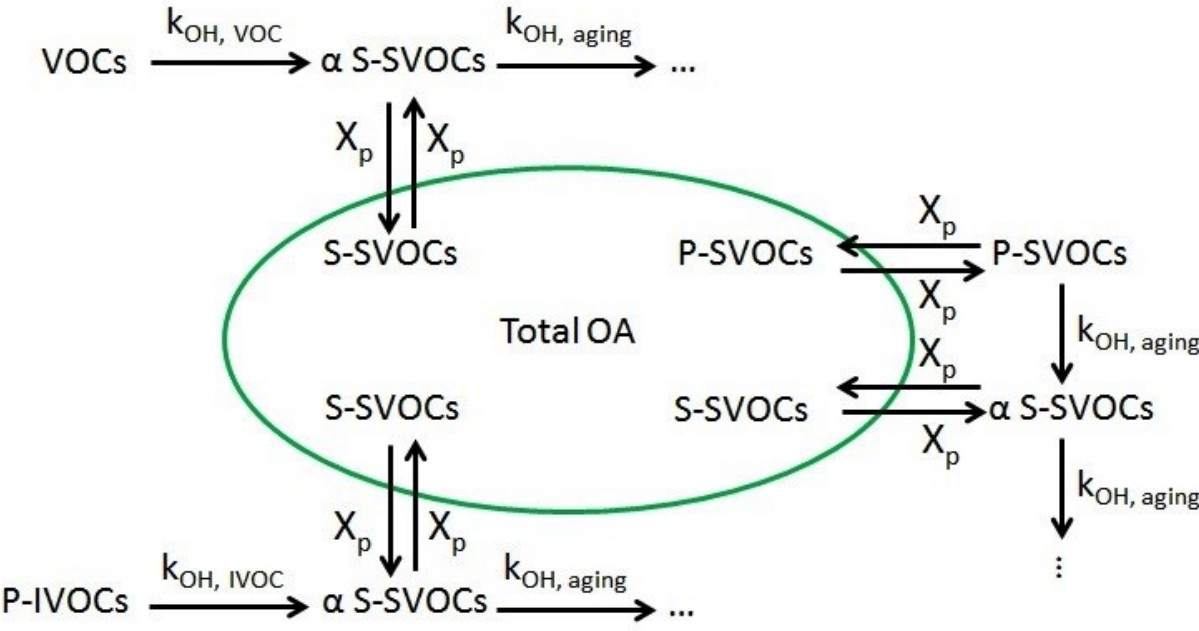


**Figure 1.** Schematic of the chemical pathways leading to the formation of SOA in the box model
where $\alpha$ is the SOA yield, $k_{OH,VOC}$ and $k_{OH,IVOC}$ are the rate constants of a VOC or an IVOC
species respectively for oxidation by OH radicals, and $X_p$ is the particle-phase fraction of a
species.

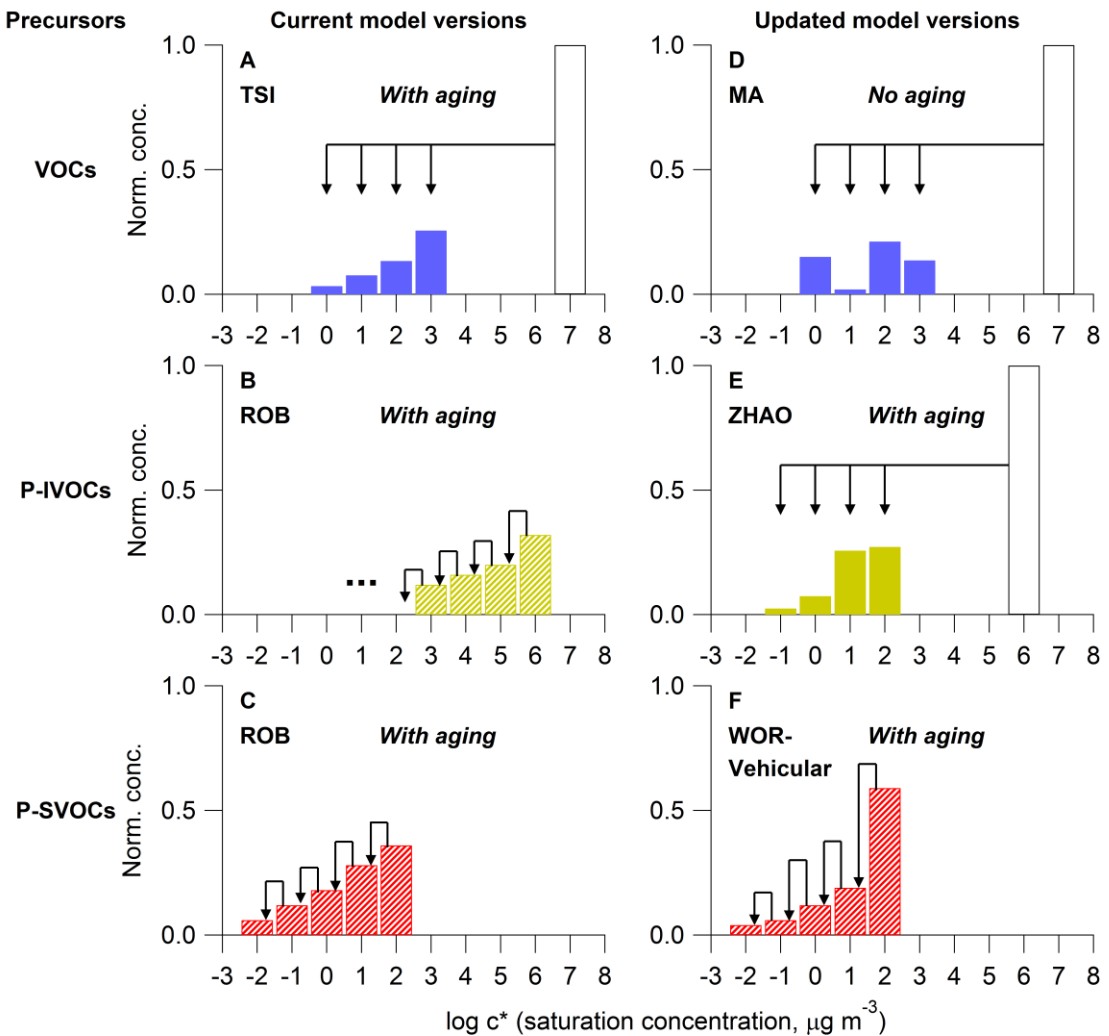


**Figure 2.** Schematic of the SOA formation parameterizations used in the model. The products formed are shown in different colors for each precursor. Note that the striped color bars indicate that the bins contain both primary and secondary organics. In panel **(A)** the parameterization of Tsimpidi et al. (2010) distributes the products of VOCs oxidation into four volatility bins. Panels **(B)** and **(C),** show the parameterization of Robinson et al. (2007) in which the volatility of the SOA precursors, specifically IVOCs and SVOCs, decrease by one order of magnitude per oxidation reaction. For P-IVOCs, aging continues to transfer mass to lower volatility bins (log c* < 2). Panel **(D)** shows the updated parameterization for VOC oxidation that accounts for gas phase wall losses, and Panel **(E)** shows the updated parameterization for P-IVOC oxidation that uses the speciated measurements of IVOCs from Zhao et al. (2014). In Panel **(F),** for the parameterization based on the measurements of Worton et al. (2014), the Robinson et al. (2007) volatility distribution is still used for the P-SVOCs emitted from cooking sources. Arrows representing the aging of SOA are omitted for clarity.

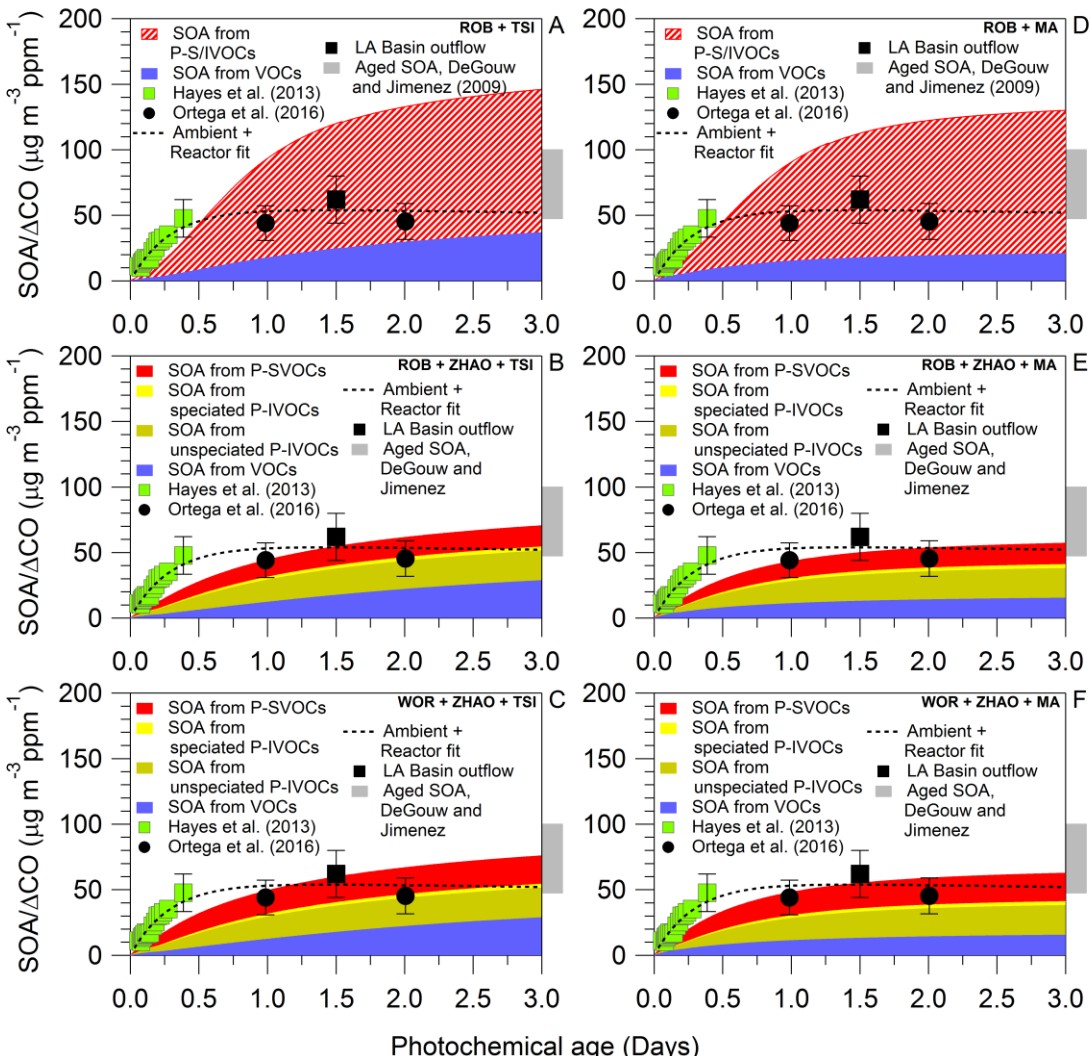

1379

**Figure 3.** Predicted urban SOA mass by all six cases for up to 3 days of photochemical aging using a reference OH radical concentration of $1.5 \times 10^6$ molec cm$^{-3}$. Background SOA is not included in the figure. The SOA concentrations have been normalized to the background subtracted CO ($\Delta$CO) concentration to account for changes in emission strengths and dilution. The SOA/$\Delta$CO data determined from the ambient and OFR measurements at Pasadena as reported by Hayes et al. (2013) (green squares) and Ortega et al. (2016) (black circles) are shown. Also shown is SOA/$\Delta$CO determined from measurements performed aboard the NOAA P3 research aircraft (black square) reported by Bahreini et al. (2012) and highly aged urban air masses (gray bar) reported by de Gouw and Jimenez (2009). The fit for ambient and reactor data reported by Ortega et al. (2016) is also shown (dotted black line).

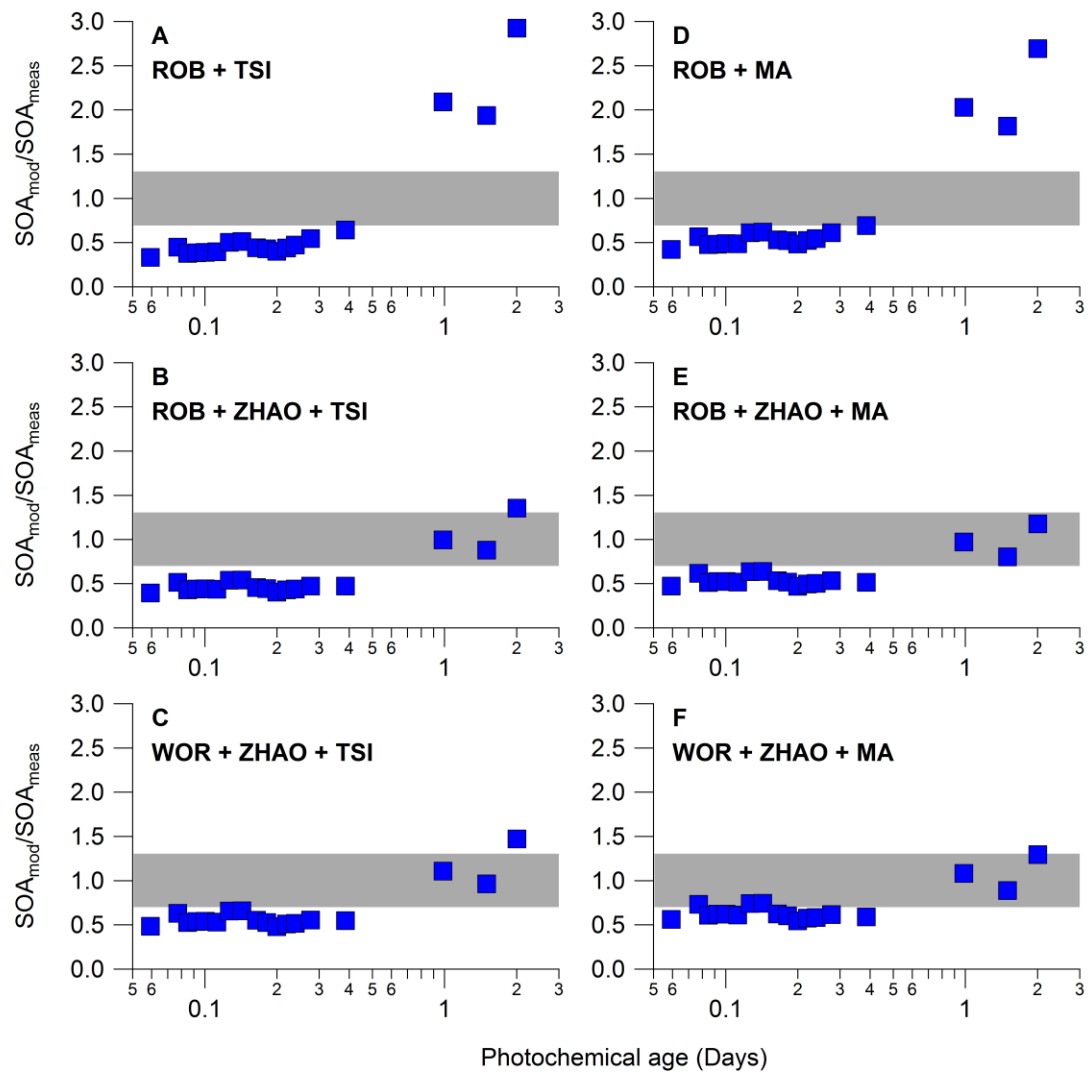

1390

Figure 4. The ratio of the modeled-to-measured SOA concentrations (blue squares) for all model cases. The measurements are the same as used in Figure 3. The gray bar indicates ratios that would correspond to model results that are within the estimated ± 30 % uncertainty of the measurements.

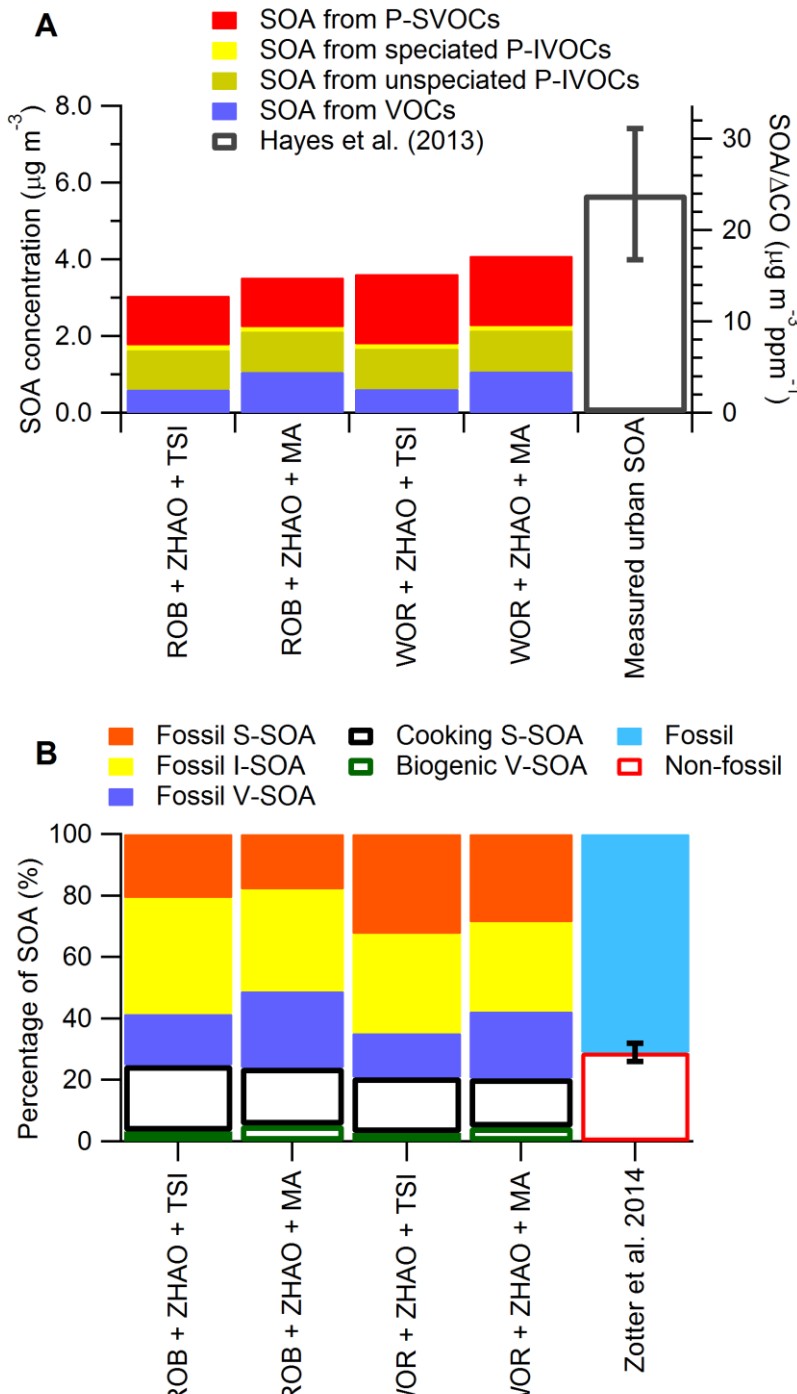

1395

1396

**Figure 5. (A)** Predicted and measured urban SOA mass for 12:00 – 15:00 local time at the Pasadena ground site. **(B)** The fractional mass of fossil S-SOA, fossil I-SOA, and fossil V-SOA, as well as cooking S-SOA and biogenic V-SOA for the same time and location. The percentage of urban SOA from fossil and non-fossil sources as reported in Zotter et al. (2014) is also displayed. The fossil sources are shown as solid bars and the non-fossil sources as hollow bars.

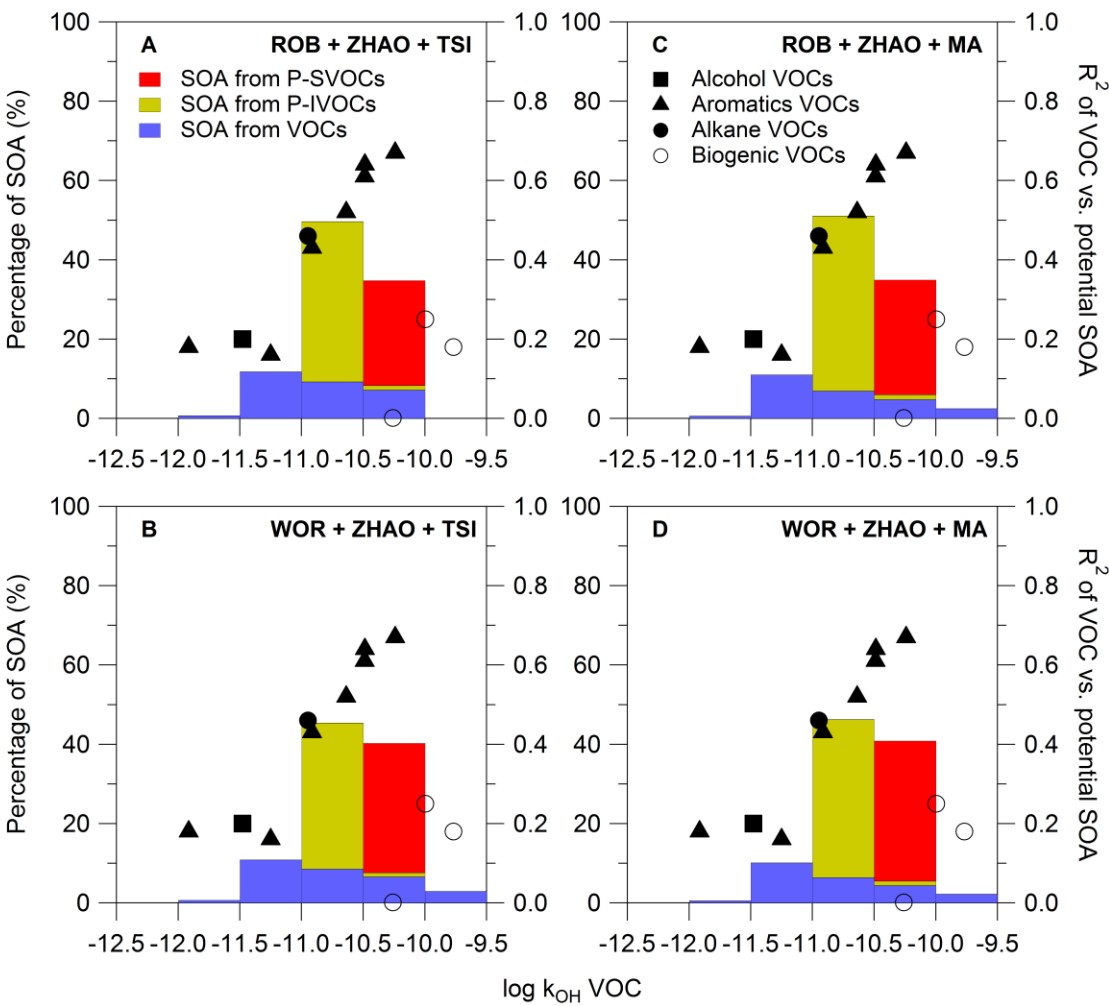

1402

**Figure 6.** Percentage of SOA mass formed from different precursors at 1.5 days of photochemical aging (at $1.5 \times 10^6$ molec OH cm$^{-3}$) binned according to precursor rate constant. The correlations ($R^2$) between the concentrations of different VOCs and the maximum SOA concentration formed in the OFR as reported by Ortega et al. (2016) are represented by the markers. The shape of the marker indicates the chemical family to which each compound belongs. For the VOCs and the P-IVOCs the rate constant is the constant for the initial oxidation reaction. The measurements of IVOCs used here allow the rate constants of these precursors to be taken from published work or estimated using structure-activity relationships as described previously (Zhao et al., 2014). For S-SOA, the rate constant is the aging rate constant reported originally by Robinson et al. (2007).

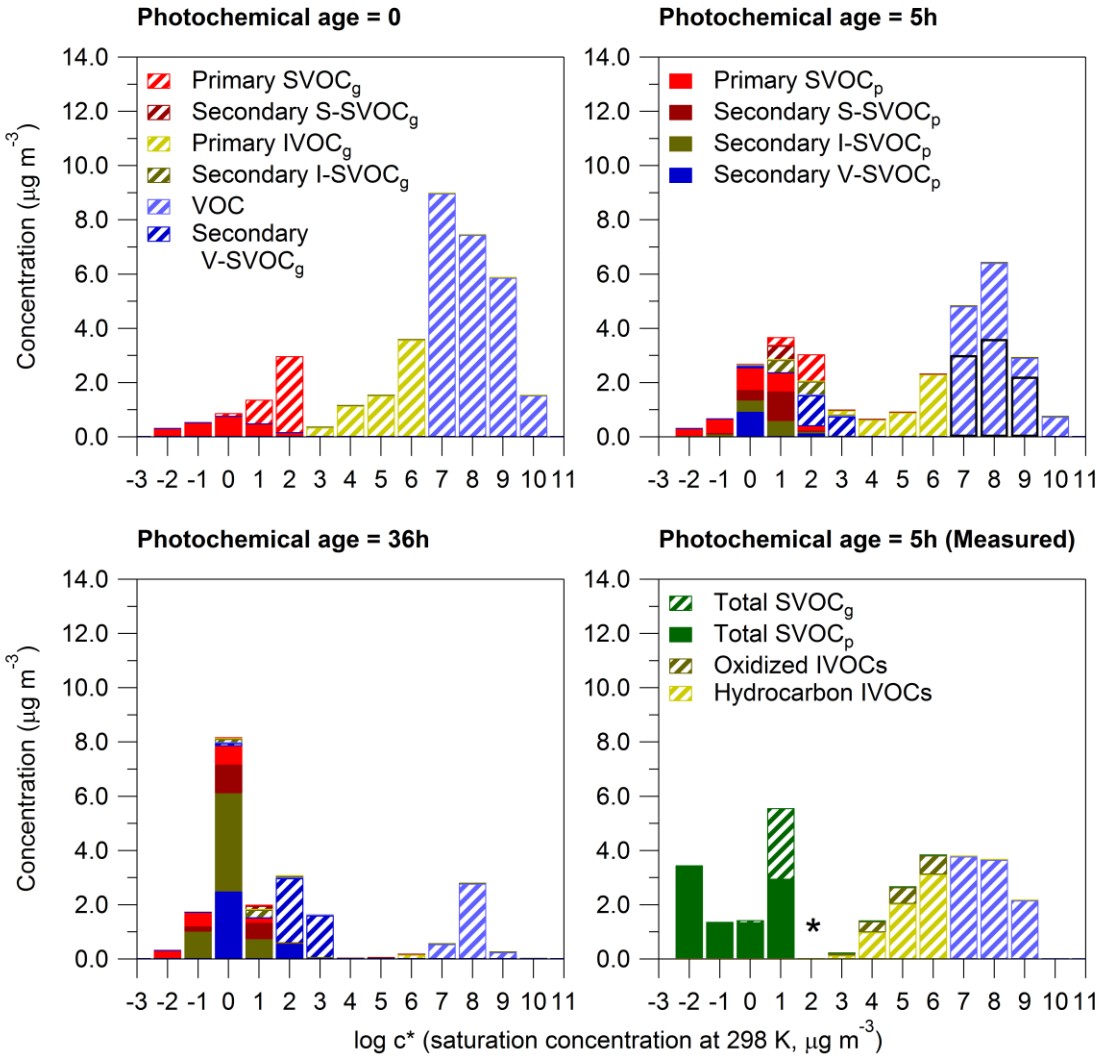

1413

**Figure 7.** OA volatility distribution as simulated by the WOR + ZHAO + MA case displayed at different photochemical ages (0, 5, and 36 h). The partitioning of the species is indicated using patterned bars for gas phase and solid bars for particle phase mass. The bottom-right graph also shows the measured volatility distribution of OA. The SVOC volatility distribution is determined using a combined thermal denuder AMS system as described in the supporting information. The IVOC volatility distribution was previously published in Zhao et al. (2014), and the VOC distribution was determined from GC-MS measurements using the SIMPOL.1 model to estimate the volatility of each VOC. The asterisk in the bin log $c^* = 2$ indicates that measurements are not available for this bin. It should be noted that not all the VOCs in the model were measured at Pasadena (see text for details). For direct visual comparison with the measurements, the simulated concentrations of only the VOCs measured at Pasadena are indicated by the black hollow bars in the bins log $c^* = 7$, 8, and 9 µg m$^{-3}$.