# Peer review of "Evaluating the impact of new observational constraints on P S/IVOC emissions, multi-generation oxidation, and chamber wall losses on SOA modeling for Los Angeles, CA"

_Atmospheric Chemistry and Physics, 2016_

## Referee Comment (RC1) · Anonymous Referee #1 · 3 Jan 2017

This paper is interesting in that new constraints on S/IVOC emissions are used together with recent VBS yield suggestions, and the results compared to a wide range of measurements. The measurement data range from near-surface to aircraft data, all evaluated using the concept of photochemical age and with SOA/CO ratios. The use of OFR data is also beneficial I think, in helping to place limits on the SOA formation and ageing-process at long photochemical ages.

Although the paper does present some interesting analysis, I think that there are some significant problems, and I cannot recommend the paper for publication until these are

addressed.

One problem is that there have been so many papers by now in which somebody identifies a problem with model-measurement discrepancies in SOA, and by tweaking the VBS parameters in some way one can get better agreement. This paper falls into that pattern, and although the authors have good reasons for their particular choice of parameter-tweaking the fact remains that there are an infinite number of ways of improving SOA predictions. The authors need to demonstrate some advantage of their schemes over others, and this requires a reliable model study.

**1 Why a box-model?**

In order to demonstrate that the merits of the tweaks used here are real, I would have wanted to see a thorough demonstration of improved model-observation performance across a range of scales. The box model used here cannot in my opinion provide such a demonstration; this study should have been conducted with a well-evaluated 3-D chemical transport model. In fact, with only four mechanisms being evaluated, and over a short period, I cannot think of any reason not to use a CTM.

Although box models are often useful for examination of, for example detailed chemical processes, or basic principles, they are not well suited to studies where comparison with ambient measurements is in focus. This has been well established for years, and is a major reason why air pollution modelling moved from the earlier EKMA-type moving-box models to 3-D models such as CAMx or CMAQ. The measurements used in this study also range from near-surface data to aircraft, which places additional constraints on the abilities of a box model. Although the authors (and those of the previous Hayes et al 2015 study which preceded this work) have put a lot of effort into the box model setup, I do not believe that any amount of effort can overcome the basic limitations of such models. Box models simply cannot account for the 3-D nature of atmospheric

dispersion, and they cannot be expected to cope with pollutant situations where non-linearity of photochemical/SOA production is expected.

The authors may argue that by scaling with CO that they remove dispersion errors but this is only partly true. This does not account for the fact that SVOC partitioning is dependent on absolute OA concentrations, and so requires simulation of e.g. urban plumes and vertical gradients.

More fundamentally, the assumption in this box-model work is that one can predict $\Delta$ VOC concentrations from $\Delta$ CO alone. There may be some merit in this for transport-derived VOC (and the VOC profiles shown in Hayes et al 2015 show surprisingly good agreement for such VOC), but such a relationship cannot hold for VOC from solvents and various production processes. Thus, we have sources of VOC and hence SOA which are not constrained. I didn't find a discussion of this in the paper.

These problems are even more difficult to deal with when comparing SOA formation at longer photochemical ages, e.g. 3 days as is done here. I would expect problems with any pollutant when running a box model over such time-scales.

The authors (also in Hayes et al 2015) do not even demonstrate that the model is capable of reproducing pollutants such as CO or NO2 reliably, and without this it is impossible to explain model-measurement discrepancies in SOA in terms of VBS parameters alone.

It was disturbing that the text didn't acknowledge these limitations, but instead all model-measurement discrepancies are assigned to VBS/SOA formation parametrizations.

Given that CTM models have already been set up and used for the CalNex campaign (Baker et al., 2015, Hayes et al., 2015, Woody et al., 2016), I would suggest that the authors re-do their work in collaboration with one of these teams.

**2 Experiment design**

I found the approach confusing in several respects.

To start with, the paragraph at the end of P9 (start of 2.2.2) is worrying. The initial POA is calculated from $\Delta$ POA/$\Delta$ CO emissions, which implies that POA is an inert pollutant. At the same time the authors use Robinson's volatility distribution to estimate all emitted P/S/IVOCs. How can this be reconciled?

At the end of 2.2.2 (P10) we read that the important ratios of IVOC or VOC to CO are derived from measurements made between 00:00 and 06:00 local time in Pasadena, 'when the amount of photochemical ageing was very low'. There are several problems here, associated with the reliability of such ratios for daytime modelling, and the assumption that other ageing processes are negligible. My guess would be that CO concentrations have a larger component of long-range transport than IVOC for example, and also that night-time chemistry would be more important than assumed here.

Thirdly, it is usually a good idea to change one aspect at a time of model simulations, in order to investigate the effect of that one change. Here though the authors move from a set of 'Tsimpidi' cases to cases where wall-loss are accounted for. At the same time they switch off the ageing of secondary SVOCs. Thus, one cannot evaluate the importance of the ageing effect alone. This would have been a useful step between the TSI and the various wall-loss cases.

Further, on p8-9, we read that ageing of secondary SVOC from 'VOC' is not included, partly because of poorly-constrained chamber data, but ageing of secondary SVOC from P-IVOCs and P-SVOCs is included for the MA cases. These choices feel rather random, and indeed seem like tweaks to give the model a decent chance of fitting the observations.

Finally, the text on p9,L301-304 anyway seems to confirm that the refit was not able to reproduce the chamber data. Although non-equilibrium reasons are given for this, I am

a little confused about the benefit of a refit that cannot reproduce the data.

**3 Yield data?**

Can the yields in Table S4 be correct? According to the manuscript (p8,L277-), this Table presents the upper limits of the SOA yields, but the numbers look rather odd compared to Table S1 which is supposed to be the lower limit. For isoprene the total yield is 0.039 in both Tables, and for other compounds the differences are sec are quite small (0.194 vs 0.200 for Ole1, 0.382 vs 0.392 for Ole2, 0.932 vs 0.939 for Aro2, 0.835 vs 0.855 for Terp). The main difference seems to be that Table S4 has mainly 0.0 for the 19 ug/m3 bin. A mistake maybe?

**4 3-day simulations?**

As noted above, I have grave reservations about the use of a box model for this study, and the extension of the simulations to 3-days in Sect. 3.1. seems hard to defend. The authors suggest that they limited the runs to 3-days to minimize the importance of missing processes such as deposition, but a box model misses all processes of dispersion, transport and even chemistry in the correct photochemical regimes when run over such a long time. I simply do not believe that such long runs with such an artificial setup can be compared with measurements in other than a superficial way.

In any case, many SVOC species will show substantial deposition over 3-days (Karl et al, 2010, Hodzic et al, 2016), as indeed would ozone and various NOy species (e.g. N2O5).

**5 Some other questions**

p3, L86 and generally. Actually VOCs are the only precursor of SOA (though of course other precursors such as NOx can be involved). VOC is a general term (defined here on L80-81) which includes SVOC and IVOC. If the authors want to use the term VOC for volatile organic compounds which are not S/IVOC then they need to refine and clarify their notation.

p3, L99. I was surprised not to see some more recent references here, since much has been done in the last years. For example, Hodzic et al 2016 seem to cover some of the same themes (wall-loss corrected VBS) as this paper, with an evaluation at global scale. Another relevant work would be that of Dunmore et al 2015 and Ots et al. 2016, where IVOC emissions from diesel were suggested to be a major source of ambient SOA.

p4, L105-107. If discussing VBS as a conceptual model, the 2-D version (Jimenez et al., 2009) deserves a mention. Actually, why was this version not used? Box/Lagrangian models have few restrictions on CPU usage, so would be a natural place to test 2-D VBS schemes.

p4, L118. loses should be losses. (There are other some small typos/English problems throughout, which should be checked.)

p6, L185-189. The text states that the potential source of error from omission if cold-starts does not apply to the total amount of vehicular POA emissions. This may be true if the absolute emission rates are not used, but surely the volatility distribution of cold-start VOC is different to that of warm-running engines?

On the same paragraph though, presumably the Worton et al data could be used to produce a new estimate of total vehicle (S/I)VOC emissions. Why wasn't this done?

p8,L276-279, 281-288. Quantify these time-scales for the reader.

[Figure]

P11, L370 and elsewhere. Define whether mass or volume fractions and stoichiometry are used. This can be an easy error, especially when the cited Donahue paper redefined Raoult's law in terms of mass rather than mole fractions.

P11, L396. 'shorter' ... than what?

P15, L519-523. Too wordy and repetitive.

P15, L531. The work of Dunmore and Ots mentioned above would support this statement.

P15, L543 on. This paragraph is a good example where the authors attribute all problems to SOA mechanisms. It may well be that the box model setup is responsible for the problems.

P17, Sect. 3.2. Given my reservations about the validity of the box-model, and its obvious lack of treatment of VOC degradation with transport time, I wasn't convinced that this section had a good basis. In addition, the manuscript is already quite long, and this section feels like a side-issue.

**6 Additional References**

Dunmore, RE et al., Diesel-related hydrocarbons can dominate gas phase reactive carbon in megacities, ACP, 2015, 15, 9983-9996

Karl, T et al., Efficient Atmospheric Cleansing of Oxidized Organic Trace Gases by Vegetation, Science, 2010, 330, 816-819

Hodzic, A et al., Rethinking the global secondary organic aerosol (SOA) budget: stronger production, faster removal, shorter lifetime, ACP, 2016, 16, 7917-7941

Ots, R et al., Simulating secondary organic aerosol from missing diesel-related

intermediate-volatility organic compound emissions during the Clean Air for London (ClearfLo) campaign, ACP, 2016, 16, 6453-6473

---

## Referee Comment (RC2) · Anonymous Referee #2 · 6 Jan 2017

Comment on "Evaluating the impact of new observational constraints on P-S/IVOC emissions, multi-generational oxidation, and chamber wall losses on SOA modeling for Los Angeles, CA" by Ma et al.

acp-2016-957

**General Comments:**

Ma and coauthors present a follow-up to the work in Hayes et al. (2015), adding more recent parameterizations for I/S-VOC emissions and yields as well as very recent approaches to correcting chamber yields for wall losses. I am satisfied with the application of the box model to the Pasadena data given the lack of quantitative statistics presented. The important limitations of the aging mechanisms and over-exuberant IVOC formation pathways is demonstrated more qualitatively than quantitatively. Also, the authors are careful to avoid strong conclusions about the dominance of SOA from IVOCs over SVOCs or vice-versa. I do urge the authors to move toward a 3D-CTM analysis in the future, particularly since I'm pretty sure input datasets exist for all the major CTMs. Although the conclusions are not exactly novel (other studies have shown that the VBS functionalization mechanisms overpredict at long photochemical lifetimes), I appreciate the demonstration of the improved parameterizations, particularly the chamber wall-loss correction. I found some aspects of the experimental design to have unnecessary limitations. Moreover, I encourage the authors to consider improving several aspects of the presentation before I recommend publication of this manuscript.

**Specific Comments:**

1) My primary question/criticism is why do the authors not investigate aging mechanisms with fragmentation given the emerging global/regional model implementations of these pathways and the low computational overhead of their own box model? This would seem like an ideal application given the experimental data available to them from CalNex. Why not use measured AMS elemental ratios to help constrain the configuration choices here? Potentially, that analysis could give complimentary information to the analysis in Section 3.2 and Fig. 6.

2) The authors discuss extensively the problems with aging VOC oxidation products and the tendency for mechanisms to accumulate mass at long photochemical lifetimes. It is important that they emphasize (stronger than they already do) that this aging approach is likely problematic precisely because it does not consider fragmentation. One of the main conclusions I read from the paper is that wall-loss corrected VOC yields should be used and aging mechanisms turned off. Conceivably, a future study will conclude that turning off aging is a bad idea because even though the OA mass is better predicted at long time, the O:C is underpredicted. The models of the future will hopefully have both more accurate yields and probably aging with both functionalization and fragmentation adequately described. To avoid confusion in the meantime, I recommend the authors refer everywhere to the TSI aging as "aging by functionalization only" or something similar, with an appropriate acronym for readability.

3) L390-394 and L480-483: The authors repeatedly refer to the OFR work of Ortega et al. (2016) to justify not including fragmentation in any model case. This argument relies on the assertion that fragmentation only played a dominant role when the OA mass began to decrease after it had plateaued for a couple of days in photochemical age space. But the OA concentrations started leveling off in that study at about 1 day. As with any competition, the manifestation of a plateau indicates to me that fragmentation is playing a role equal to that of functionalization. So sentences like L477-480 and L482-483 are pretty confusing, if not misleading.

4) The application of the wall-loss corrected chamber yields seems problematic to me. First of all, many of the studies used to inform the Tsimpidi et al. (2010) yield set included seed aerosol in their experiment. As the authors point out multiple times, the data they have included in Table S4 should be considered an upper bound. However, I fear that their demonstration of this approach will encourage others to blanket apply the parameters of Krechmer et al. (2016) to historical chamber yields without considering the details and possible interferences. I encourage the authors to describe in detail the problems with applying the narrowly defined Krechmer $C_w$'s to existing data and repeat that paper's call for more detailed analysis of chamber data before the community gobbles this simple approach and then moves on to the next hot SOA formation topic.

5) Why is the SOA mass in Fig. 5 not also divided by CO concentrations to correct for dilution?

6) L257-258: Woody et al. (2016) proposed a meat cooking volatility distribution. Why not try this one in a sensitivity test?

7) Can the authors clarify more directly why the model with the Worton parameters for SVOCs gives more OA than that with the Robinson parameters? The volatility distribution and Fig. 2 show pretty clearly that the emissions are substantially higher in volatility. Is the difference really from the added 7.5% mass that comes with the 1-bin aging mechanism? If that's the case, please emphasize more clearly the uncertainty in this parameter/approach to put the differences in these two model runs into context.

8) Section 3.2 and Fig. 6: This analysis is an interesting idea but I don't think the slight differences among the model cases warrant such a long-winded discussion and detailed figure. It would be enough to add a comment to section 3.1 that the WOR cases give more SOA from precursors with $k_{OH}$ in the range identified by Ortega et al. (2016). The abstract and conclusions would need to be correspondingly reduced.

9) L716-718 and L796-797: Why do the authors not discuss the limitations of their aging mechanisms that only reduce volatility by one bin at a time? It is possible that a compound can shift more than one generation in volatility upon oxidation; the more recent 2D-VBS approaches and the SOM methods allow for multi-decadal shifts in volatility. Approaches like these might push the products below the "oxidation-partitioning barrier" manifested when compounds are protected from gas-phase oxidation.

10) L760-762: How do these reaction rate constants compare to estimation methods developed for the 2D-VBS? If you used those approximations (based on $C^*$ and assumed O:C) would you do better?

**Minor Changes/Typos:**

1) L48-50: This sentence should say something about how the two methods predict similar mass at short to moderate photochemical ages.

2) L82: Consider replacing "nucleate" with "form".

3) L742-743: Make sure to also mention that Woody et al. (2016) cited excessive model dispersion as a potential complicating factor.

4) I recommend adding figures with SOA mass curves (not just the yields) for each of the VOC product species to the supporting information, thereby visually demonstrating the effects of the upper- and lower-bound yield parameterizations. It would be a good idea to assume a background concentration equal to 2.1 ug m-3 (or greater if you just want to take an average of your total OA, model wide) like in the model so that you get relevant partitioning.

5) SI, L6-22: I found the derivation of the wall-loss correction confusing. First, the quantity in parentheses in equations 5 and 6 should be the reciprocal. I assume they used the correct form for the calculation because I calculated the adjusted ARO2 and it would have been way off using the equation as it is written. Also, [VOC] should be replaced with something more accurate like [ΔROG] or [ΔVOC]. It would be helpful to explain briefly why the mass of compounds on the walls, $C_w$, is a function of $C^*$ alone and not $C_{tot}$, $C_g$ or $C_p$. This is essentially a consequence of the equilibrium assumption in the chamber analysis, as I understand it.

---

## Author Response (AR1)

**Key:** Solid Blue = Responses, *Italicized blue = new text,* Quotation marks = new and existing text from the manuscript.

We would like to thank the reviewers for taking the time to review our paper and for their thoughtful comments. Their comments have helped us clarify and improve the manuscript. We have reproduced the reviewer comments in black text. For ease of review, our responses are given in blue text, while the new text added to the manuscript is given in blue Italics and the original text from the submitted manuscript remains un-italicized. We would also like to point out that the numbering of the figures from the revised manuscript is used here in the responses and that the figures only used for responses are noted with the prefix ''R''.

**Referee 1 Comments**

**R1.1.** This paper is interesting in that new constraints on S/IVOC emissions are used together with recent VBS yield suggestions, and the results compared to a wide range of measurements. The measurement data range from near-surface to aircraft data, all evaluated using the concept of photochemical age and with SOA/CO ratios. The use of OFR data is also beneficial I think, in helping to place limits on the SOA formation and ageing-process at long photochemical ages.

Although the paper does present some interesting analysis, I think that there are some significant problems, and I cannot recommend the paper for publication until these are addressed. One problem is that there have been so many papers by now in which somebody identifies a problem with model-measurement discrepancies in SOA, and by tweaking the VBS parameters in some way one can get better agreement. This paper falls into that pattern, and although the authors have good reasons for their particular choice of parameter-tweaking the fact remains that there are an infinite number of ways of improving SOA predictions. The authors need to demonstrate some advantage of their schemes over others, and this requires a reliable model study.

We address to this comment by kindly referring the reviewer to the following paragraph written in the submitted manuscript (p. 5, L152) and by adding some sentences (in *Italics*) for clarity:

> **"The goal of this study is to use several recently published results to better evaluate and constrain the box model introduced in our previous work, and thus facilitate the identification of parameterizations that can be eventually incorporated into 3-D air quality models to accurately predict SOA for the right reasons.** *It is important to note that parameterizations used in the box model are based on several published measurements taken from laboratory experiments and field studies that provide more realistic constraints than in previous versions and that were not available to be implemented in Hayes et al. (2015).* **In particular, our work here improves the box model by incorporating recently published measurements of P-IVOCs and P-SVOCs that allow better constraining of the concentration, reactivity, yields, and volatility of these precursors (Worton et al., 2014; Zhao et al., 2014). In addition, given that experiments in environmental chambers may underestimate SOA yields for the VOCs due to losses of semi-volatile gases to the chamber walls (Zhang et al., 2014), the SOA yields from VOCs have been re-estimated using a very recent**

**parameterization of these wall-losses (Krechmer et al., 2016). The wall-loss corrected yields obtained**
**are then used in the model in a sensitivity study to evaluate the corresponding change in the modeled**
**SOA concentrations.** *The model is modified based on these literature constraints. No model tuning is*
*performed with the goal of improving the agreement with the observations."*

We also want to clarify that, contrary to the statement from the reviewer, no model tuning is performed
in our work at all. That is to say, we test SOA models and parameters based on the literature, and we do
not derive new versions based on fitting the observations. The results are obtained by directly
incorporating into the model the results mentioned above from each study (P-IVOC concentrations,
volatility, etc.) with no *a priori* knowledge that those model cases would have better (or worse)
model/measurement agreement. In other words, in the development of the model cases
model/measurement agreement with respect to SOA concentration was not used to determine the
model parameters and the parameters in each case are not "tuned".

**Why a box-model?**

**R1.1.1** In order to demonstrate that the merits of the tweaks used here are real, I would have wanted to
see a thorough demonstration of improved model-observation performance across a range of scales.
The box model used here cannot in my opinion provide such a demonstration; this study should have
been conducted with a well-evaluated 3-D chemical transport model. In fact, with only four mechanisms
being evaluated, and over a short period, I cannot think of any reason not to use a CTM.

Although box models are often useful for examination of, for example detailed chemical processes, or
basic principles, they are not well suited to studies where comparison with ambient measurements is in
focus. This has been well established for years, and is a major reason why air pollution modelling moved
from the earlier EKMA-type moving box models to 3-D models such as CAMx or CMAQ. The
measurements used in this study also range from near-surface data to aircraft, which places additional
constraints on the abilities of a box model. Although the authors (and those of the previous Hayes et al
2015 study which preceded this work) have put a lot of effort into the box model setup, I do not believe
that any amount of effort can overcome the basic limitations of such models. Box models simply cannot
account for the 3-D nature of atmospheric dispersion, and they cannot be expected to cope with
pollutant situations where nonlinearity of photochemical/SOA production is expected.

We very strongly disagree with these statements about 3-D models always being superior to box
models, and in fact think that the opposite is the case in some cases, as already documented by many
prominent papers in the literature. We address to this comment by kindly referring the referee to the
following paragraph written in the submitted manuscript (p. 4, L138) and by adding some sentences (in
*Italic*) for clarity:

[revised manuscript text omitted]

**R1.1.2** The authors may argue that by scaling with CO that they remove dispersion errors but this is only partly true. This does not account for the fact that SVOC partitioning is dependent on absolute OA concentrations, and so requires simulation of e.g. urban plumes and vertical gradients.

This is a small effect, which we had already addressed previously with a sensitivity study in Dzepina et al. (2009). We address this comment by adding the following section in the text:

*Section 2.4 Correction for changes in partitioning due to emissions into a shallower boundary layer upwind of Pasadena*

*"As described in Hayes et al. (2015), during the transport of the pollutants to Pasadena, the planetary boundary layer (PBL) heights increase during the day. Using CO as a conservative tracer of emissions does not account for how the shallow boundary layer over Los Angeles in the morning influences gas-particle partitioning due to lower vertical mixing and higher absolute POA and SOA concentrations at that time. Thus, as shown in the gas-particle partitioning equation above, there will be a higher partitioning of the species to the particle phase and less gas-phase oxidation of primary and secondary SVOCs. Later in the morning and into the afternoon the PBL height increases (Hayes et al. 2013) diluting the POA and urban SOA mass as photochemical ages increases. However this is a relatively small effect as the partitioning calculation in the SOA model is relatively insensitive to this effect and the absolute OA concentrations (Dzepina et al., 2009; Hayes et al., 2015). Our previous work (Hayes et al., 2015) found in a sensitivity study a +4/-12% variation in predicted urban SOA when various limiting cases were explored for simulation of the PBL (e.g. immediate dilution to the maximum PBL height measured in Pasadena versus a gradual increase during the morning).*

*To account for the effect of absolute OA mass on the partitioning calculation, the absolute partitioning mass is corrected using the following method. A PBL height of 345 m is used for a photochemical age of 0 h and it reaches a height 855 m at a photochemical age of 9.2 h, which is the maximum age for the ambient field data. Between the two points, the PBL is assumed to increase linearly. The boundary layer heights are determined using ceilometer measurements from Pasadena at 6:00 - 9:00 and 12:00 - 15:00 local time, respectively (Hayes et al. 2013). The second period is chosen because it corresponds to when the maximum photochemical age is observed at the site. The first period is chosen based on transport times calculated for the plume from downtown Los Angeles (Washenfelder et al. 2011) that arrives in Pasadena during the afternoon. There are certain limitations to this correction for the partitioning calculation. First, the correction is based on a conceptual framework in which a plume is emitted and then transported to Pasadena without further addition of POA or SOA precursors. A second limitation is that we do not account for further dilution that may occur as the plume is advected downwind of Pasadena. However, such dilution is not pertinent to the OFR measurements, and so for photochemical ages beyond ambient levels observed at Pasadena, we focus our analysis on the comparison with the OFR measurements."*

**R1.1.3** More fundamentally, the assumption in this box-model work is that one can predict ΔVOC
concentrations from ΔCO alone. There may be some merit in this for transport derived VOC (and the
VOC profiles shown in Hayes et al 2015 show surprisingly good agreement for such VOC), but such a
relationship cannot hold for VOC from solvents and various production processes. Thus, we have
sources of VOC and hence SOA which are not constrained. I didn't find a discussion of this in the paper.

We address to this comment by adding the following paragraph in the experimental section.

*''It should be noted that the use of VOC emission ratios to CO to estimate VOC emissions does*
*not assume that VOCs are always co-emitted with CO. Rather, it assumes that VOC emission sources*
*are individually small and finely dispersed in an urban area, so that they are spatially intermingled*
*with the sources of CO. Moreover, previous studies have measured the emission ratios of*
*anthropogenic VOCs with respect to CO and the results show that vehicle exhaust is a major source of*
*VOC and CO (Borbon et al. 2013, Warneke et al. 2007). Furthermore, the ratios are consistent both*
*temporally and spatially. Thus, when thinking of the entire urban area as a source, the use of emission*
*ratios to CO is justified. As shown in Hayes et al. (2015) in the supporting information, the modeled*
*VOC concentrations are consistent with the measurements indicating that major VOCs sources have*
*not been omitted, and the smooth time variations of the VOC concentrations support the use of a*
*"global urban source".''*

**R1.1.4** These problems are even more difficult to deal with when comparing SOA formation at longer
photochemical ages, e.g. 3 days as is done here. I would expect problems with any pollutant when
running a box model over such time-scales.

We address to this comment by adding the following paragraph in section 3.1 of the submitted
manuscript:

*''As displayed in the graphs for Fig. 3, it should be noted the measurements from the OFR*
*(Ortega et al. 2016) and from the NOAA P3 research aircraft (Bahreini et al. 2012) give quite similar*
*results for SOA/ΔCO. The OFR measurements are not affected by particle deposition that would occur*
*in the atmosphere at long timescales or photochemical ages. Only a few percent of the particles are*
*lost to the walls of the reactor, and this process has been corrected for already in the results of Ortega*
*et al. The similarity in the two types of observations suggests that ambient particle deposition and*
*plume dispersion do not significantly change the SOA/ΔCO ratio over the photochemical ages*
*analyzed here.''*

**R1.1.5** The authors (also in Hayes et al. 2015) do not even demonstrate that the model is capable of
reproducing pollutants such as CO or NO2 reliably, and without this it is impossible to explain model-
measurement discrepancies in SOA in terms of VBS parameters alone.

It was disturbing that the text didn't acknowledge these limitations, but instead all model-measurement
discrepancies are assigned to VBS/SOA formation parametrizations.

As the $NO_X$ levels and CO are constraints already used in the model set-up, it would not be meaningful to
perform the diagnostics described by the reviewer, although we certainly agree that those comparisons
would be worthwhile for (unconstrained) 3-D models. The ability to constrain important parameters is
one key advantage of constrained box models for this type of application. To better explain how the
model set-up is evaluated we have added the following text in section 2.2 of the manuscript.

*"It should be noted that the model uses CO and $NO_x$ as inputs to constrain the model and the*
*SOA yields for high-$NO_X$ conditions are used, based on our previous work (Hayes et al. 2013; 2015).*
*Therefore, to verify model performance both predictions of VOC and POA concentrations have been*
*compared against field measurements and the model performance appears to be satisfactory (Hayes*
*et al. 2015)."*

We agree with the reviewer that it is useful if the conclusions drawn from this study more clearly
acknowledge the model uncertainties, which appear to be confusing in the submitted version.
Therefore, we have updated the text as described in our response to comment R1.5.12 below.

**R1.1.6** Given that CTM models have already been set up and used for the CalNex campaign (Baker et al.,
2015, Hayes et al., 2015, Woody et al., 2016), I would suggest that the authors re-do their work in
collaboration with one of these teams.

Based on our points described above, especially R1.1.1, it seems reasonable to conclude that a box
model provides important scientific value that complements 3-D models and is superior to those models
for some scientific questions. Indeed, we have contributed to both box modeling as well as collaborated
closely in several 3-D modeling studies for CalNex. From those experiences, we have concluded that the
box model can be superior to a 3-D model for this application, as it eliminates or greatly reduces many
potential errors in, e.g., the photochemical age, dispersion, and emissions, by the use of constraints. We
are not saying that box models can always provide a comparable alternative to 3-D models to study
chemical processes in all cases, but this is clearly the case when source regions and transport are simple
and well-characterized as in this study.

**2 Experiment design**

**R1.2.1** I found the approach confusing in several respects.

To start with, the paragraph at the end of P9 (start of 2.2.2) is worrying. The initial POA is calculated from ΔPOA/ΔCO emissions, which implies that POA is an inert pollutant. At the same time the authors use Robinson's volatility distribution to estimate all emitted P/S/IVOCs. How can this be reconciled?

This topic has been discussed extensively in previous work (e.g. Dzepina et al., 2009, 2011; Hayes et al., 2015). We address this comment by updating the text below in the submitted manuscript (p. 9, L324).

*''The initial POA concentration is determined from the product of the background-subtracted CO concentration and the ΔPOA/ΔCO emission ratio (Hayes et al., 2015). While this ratio may change due to evaporation/condensation or photochemical oxidation of POA, our previous work (Hayes et al. 2013) has shown that ΔPOA/ΔCO does not change significantly at the Pasadena ground site with observed photochemical age indicating that the ratio is insensitive to the extent of photochemical oxidation. Furthermore, it was calculated that the ratio would increase by 28% for an increase of OA concentration from 5 to 15 μg m$^{-3}$, concentrations that are representative of this study. This possible source of error is substantially smaller than current errors suggested for P-S/IVOC emission inventories in 3-D models, where current schemes are based on scaling POA emission inventories with scaling factors that are not well constrained (Woody et al. 2016).''*

**R1.2.2** At the end of 2.2.2 (P10) we read that the important ratios of IVOC or VOC to CO are derived from measurements made between 00:00 and 06:00 local time in Pasadena, 'when the amount of photochemical ageing was very low'. There are several problems here, associated with the reliability of such ratios for daytime modelling, and the assumption that other ageing processes are negligible. My guess would be that CO concentrations have a larger component of long-range transport than IVOC for example, and also that night-time chemistry would be more important than assumed here.

CO is well-known to have a background from long-range transport, which is estimated and always subtracted before taking ratios, as described in Hayes et al. (2013) and references therein. Thus ΔCO represents the urban contribution to CO and is the appropriate quantity to use here. For clarity, we update the sentence below in the submitted manuscript (p. 10, L345-346):

*"During the regression analyses the x-intercept was fixed at 105 ppbv CO to account for the background concentration of CO determined in our previous work (Hayes et al., 2013). Thus, the slope of the resulting line corresponds to the estimated emission ratio (ΔIVOC/ΔCO)."*

We further address the potential uncertainty in IVOCs suggested by this comment by running a sensitivity case in which the IVOC initial concentration is calculated using the observed photochemical age, the IVOC measurements at Pasadena, and the estimated IVOC oxidation rate constants following Zhao et al. (2014). This alternate approach does not rely on the nighttime ratios of IVOC to CO.

For comparison, we also include here the SOA formation results before running this sensitivity study
that are in the manuscript.

[Figure]

Figure 3. Predicted urban SOA mass for the E) ROB + ZHAO + MA and F) WOR + ZHAO + MA cases with
the original model set-up for this work.

The figure below represents the SOA formation for the same two cases as above but for initial IVOC
concentrations calculated without using the IVOC to CO ratios (as described above).

[Figure]

Figure S10. Predicted urban SOA mass for the A) ROB + ZHAO + MA and B) WOR + ZHAO + MA cases
when using IVOC initial concentrations determined using photochemical age, the Pasadena IVOC
concentrations, and the estimated IVOC oxidation rate constants.

To summarize these findings, we have added the paragraph below in the manuscript in section 3.3 and
included Figure S10 in the supporting information.

*''To further explore the impact of potential errors in the initial IVOC concentrations, a*
*sensitivity study has been carried out using initial concentrations calculated based on the observed*
*photochemical age and measured IVOC concentrations at Pasadena as well as the estimated IVOC*
*oxidation rate constants (Zhao et al., 2014). This alternate approach is implemented for the ROB +*
*ZHAO + MA and WOR + ZHAO + MA cases and does not use nighttime IVOC-to-CO ratios. The results*
*when using this alternative approach are shown in the supporting information (Figure S10). When*
*comparing Fig. S10 with Fig. 3, differences are minor. The model/measurement agreement improves*
*slightly at shorter photochemical ages (less than 1 day). At the same time a slightly larger over-*
*prediction is observed at longer photochemical ages. However, the formation of SOA modeled in this*
*sensitivity test is similar to the original cases from Fig. 3 with an average difference of only 21 %,*
*which represent a relatively small error compared to other uncertainties in SOA modeling. The IVOC*
*initial concentrations used in this sensitivity test are slightly higher than those calculated using the*
*IVOC-to-CO ratio, which explain the small increase of modeled SOA/ΔCO. Ultimately, the different*
*approaches for determining the initial IVOC concentration in the model are reasonably consistent, and*
*both approaches perform similarly given the model and measurement uncertainties.''*

**R1.2.3** Thirdly, it is usually a good idea to change one aspect at a time of model simulations, in order to
investigate the effect of that one change. Here though the authors move from a set of 'Tsimpidi' cases
to cases where wall-loss are accounted for. At the same time they switch off the ageing of secondary
SVOCs. Thus, one cannot evaluate the importance of the ageing effect alone. This would have been a
useful step between the TSI and the various wall-loss cases.

We address this comment by kindly referring the reviewer to the updated texts below, which are copied
from the revised manuscript (page and line numbers written next to each section) where we discuss the
inclusion of the ''aging'' mechanisms in the model.

**From the Introduction (text added at p. 4, L125 in original manuscript):**

*''These "aging" mechanisms increase VOC yields to levels much higher than those observed in*
*chamber studies since it was perceived that the yields may be too low in chambers compared to the*
*real atmosphere. The "aging" mechanisms were added to chamber yields that were obtained without*
*using aging as part of the fits of the chamber data. In some model applications they improve model*
*agreement with field measurements (Ahmadov et al., 2012), while at long photochemical ages they*
*lead to model SOA formation that is substantially larger than observed (e.g. Hayes et al., 2015;*
*Dzepina et al., 2011).''*

**From section 2.2 (text added at p. 8, L289 in original manuscript):**

*''The three model cases accounting for wall losses of organic vapors are named ROB + MA,*
*ROB + ZHAO + MA, and WOR + ZHAO + MA. For these cases, the aging of the secondary SVOCs formed*
*from the oxidation of VOCs was not included, since multi-generation oxidation is not well-constrained*

**301**   **using data from chamber studies that are run over relatively short time-scales (i.e. hours). In addition,**
**302**   **aging and correcting for wall-losses of organic vapors have been separately proposed to close the gap**
**303**   **between observed and predicted SOA concentration from pre-2007 models*, and are thought to***
**304**   ***represent the same "missing SOA mass." Therefore, we run the model with one of these options at a***
**305**   ***time, as they are conceptually different representations of the same phenomenology.* The aging of**
**306**   **secondary SVOCs formed from the oxidation of P-IVOCs (and P-SVOCs) has been kept for all of the MA**
**307**   **cases, however. *To our knowledge, P-IVOC and P-SVOC mechanisms proposed in the literature have***
**308**   ***always included aging.''***

**309**   As discussed in the submitted manuscript, the use of "aging" mechanisms were introduced to represent
**310**   processes that increase SOA yields in the real world compared to chambers, and that are now known to
**311**   be mostly due to vapor losses to chamber walls.

**312**   **R1.2.4** Further, on p8-9, we read that ageing of secondary SVOC from 'VOC' is not included, partly
**313**   because of poorly-constrained chamber data, but ageing of secondary SVOC from P-IVOCs and P-SVOCs
**314**   is included for the MA cases. These choices feel rather random, and indeed seem like tweaks to give the
**315**   model a decent chance of fitting the observations.

**316**   We address to this comment by clarifying that the choices are not random and we kindly refer the
**317**   reviewer to our response in R1.2.3. As discussed in our previous response, the VOC aging is conceptually
**318**   replaced by the correction for vapor wall losses on chamber walls. Therefore our choices are self-
**319**   consistent, and they are the simplest choices that can be made based on the literature.

**320**   **R1.2.5** Finally, the text on p9, L301-304 anyway seems to confirm that the refit was not able to
**321**   reproduce the chamber data. Although non-equilibrium reasons are given for this, I am a little confused
**322**   about the benefit of a refit that cannot reproduce the data.

**323**   To clarify, the refit *was able* to reproduce chamber data very well for the oxidation of VOCs.  Therefore,
**324**   it seems reasonable to conclude that refitting the data for the VOCs is beneficial, since wall-loses appear
**325**   to be an important process that should be accounted for as best as possible. The refitting procedure was
**326**   unsuccessful only in the case of the IVOCs. We have updated the text to more clearly explain these
**327**   results.

**328**         ***"Indeed, when trying to refit the VOC and IVOC yield curves, the model assuming equilibrium***
**329**   ***partitioning between particles, the gas phase, and the walls was able to reproduce the yield curves for***
**330**   ***VOCs, but not for IVOCs. This difference in the results is consistent with equilibrium not having been***
**331**   ***reached during the chamber studies on the IVOCs, which produce a greater amount of lower volatility***
**332**   ***SVOCs when compared to VOCs during oxidation. These lower volatility SVOCs have relatively slow***
**333**   ***evaporation rates from the particles, which prevents the chamber system from reaching equilibrium***
**334**   ***(Ye et al. 2016).''***

**335**

**Yield data?**

**R1.3.** Can the yields in Table S4 be correct? According to the manuscript (p8,L277-), this Table presents the upper limits of the SOA yields, but the numbers look rather odd compared to Table S1 which is supposed to be the lower limit. For isoprene the total yield is 0.039 in both Tables, and for other compounds the differences are sec are quite small (0.194 vs 0.200 for Ole1, 0.382 vs 0.392 for Ole2, 0.932 vs 0.939 for Aro2, 0.835 vs 0.855 for Terp). The main difference seems to be that Table S4 has mainly 0.0 for the 19 ug/m3 bin. A mistake maybe?

Firstly, we confirm that the SOA yields in Table S4 are indeed correct given our methodology.

Second, it is important to distinguish between total SOA yield , Y (particle-phase only), and the lumped SVOC yields, $\alpha_i$ (gas plus particle phases in a single volatility bin), which are related by the equation below (in the absence of wall-losses), and where $C_{OA}$ is the particle concentration.

$$Y = \alpha_1 \left(1 + \frac{1}{C_{OA}}\right)^{-1} + \alpha_{10} \left(1 + \frac{10}{C_{OA}}\right)^{-1} + \alpha_{100} \left(1 + \frac{100}{C_{OA}}\right)^{-1} + \alpha_{1000} \left(1 + \frac{1000}{C_{OA}}\right)^{-1} \quad (1)$$

The equation above can be modified as discussed in our manuscript to include the partitioning of the organics to the wall where $C_W$ is the effective wall mass concentration.

$$Y = \alpha_1 \left(1 + \frac{1}{C_{OA}} + \frac{C_{w,1}}{C_{OA}}\right)^{-1} + \alpha_{10} \left(1 + \frac{10}{C_{OA}} + \frac{C_{w,10}}{C_{OA}}\right)^{-1}$$

$$+ \alpha_{100} \left(1 + \frac{100}{C_{OA}} + \frac{C_{w,100}}{C_{OA}}\right)^{-1} + \alpha_{1000} \left(1 + \frac{1000}{C_{OA}} + \frac{C_{w,1000}}{C_{OA}}\right)^{-1} \quad (2)$$

According to equations 1 and 2, at low $C_{OA}$ the observed Y will be lower than that observed in the absence of wall losses. On the other hand, when $C_{OA}$ is much higher than $C_W$, the term $C_{w,i}/C_{OA}$ converges to 0 and equation 2 becomes identical to equation 1. Furthermore, at very high $C_{OA}$, Y is simply the sum of the $\alpha_i$ values. Therefore, Y at very high $C_{OA}$ concentrations is the same with or without wall losses and thus the sum of $\alpha_i$ is also the same with or without wall loses. Therefore, the observation that the total SVOC yields are quite similar between Table S1 and S4 is not surprising, and actually expected. However, the difference in the volatility distribution of the yields, with a shift towards lower volatility when wall losses of organic vapors are accounted for, means that Y will be higher for low OA concentrations (typical of ambient conditions) and thus OA will have a tendency to form faster at low photochemical aging.

To clarify this point we have added the following text to the manuscript in section 2.2.

*''Furthermore, as described in the supporting information, the updated SOA yields for VOC*
*oxidation result in distribution of SVOC mass into lower volatility bins compared to the original*
*parameterization, although the sum for the SVOC yields ($\alpha_i$) remains similar. In the absence of aging,*
*the SOA yields, Y, resulting from the wall-loss correction should be considered upper limits (MA*
*parameterization), whereas the original yields serve as lower limits due to the considerations*
*discussed above (TSI parameterization without aging). As shown in the supporting information*
*(Figures S1 - S7) when aging (TSI parameterization with aging) is included the SOA yields increase*
*beyond those observed when applying the wall loss correction for most of the VOC classes at longer*
*photochemical ages. (It should be noted that SOA masses in Figues S1 – S7 were calculated using the*
*same background as for the other model cases, 2.1 $\mu$g m$^{-3}$.) This feature of the aging parameterization*
*is likely to blame for SOA over-predictions observed at long aging times when comparing with*
*ambient data (e.g. Dzepina et al., 2011; Hayes et al., 2015).''*

**3-day simulations?**

**R1.4.** As noted above, I have grave reservations about the use of a box model for this study, and the
extension of the simulations to 3-days in Sect. 3.1. seems hard to defend. The authors suggest that they
limited the runs to 3-days to minimize the importance of missing processes such as deposition, but a box
model misses all processes of dispersion, transport and even chemistry in the correct photochemical
regimes when run over such a long time. I simply do not believe that such long runs with such an
artificial setup can be compared with measurements in other than a superficial way.

In any case, many SVOC species will show substantial deposition over 3-days (Karl et al, 2010, Hodzic et
al, 2016), as indeed would ozone and various NOy species (e.g. N2O5).

We refer the reviewer to our response to comment R1.1.4. In addition, we have clarified the manuscript
in order to focus on the comparison of our results with the OFR measurements, which are completely
consistent with the model set-up, where deposition is also not important. We note that the box model is
run for the conditions of the OFR itself, which is not problematic. The consistency of the OFR and aircraft
results indicates that deposition does not have a major influence on the model results over short
timescales.

**Some other questions**

**R1.5.1** p3, L86 and generally. Actually VOCs are the only precursor of SOA (though of course other precursors such as NOx can be involved). VOC is a general term (defined here on L80-81) which includes SVOC and IVOC. If the authors want to use the term VOC for volatile organic compounds which are not S/IVOC then they need to refine and clarify their notation.

This is a semantic difference. In our work we use the term VOC as separate from S/IVOCs, while other authors (and reviewer 1) include S/IVOCs as part of the term VOCs. Different definitions are often used in the scientific literature for many terms, which is fine as long as each paper is clear on which definition is used. Thus, we address this comment by adding the following sentences in the introduction section.

*"The notation used when discussing SOA precursors in this paper is similar to Hayes et al. (2015). We differentiate VOCs, IVOCs and SVOCs by their effective saturation concentration (c\*). Therefore, SVOCs and IVOCs have volatilities ranging from $c* = 10^{-2}$ to $10^2$ and $10^3$ to $10^6$ µg m$^{-3}$ respectively, while VOCs are in the bins of $c* \geq 10^7$ µg m$^{-3}$."*

**R1.5.2** p3, L99. I was surprised not to see some more recent references here, since much has been done in the last years. For example, Hodzic et al 2016 seem to cover some of the same themes (wall-loss corrected VBS) as this paper, with an evaluation at global scale. Another relevant work would be that of Dunmore et al 2015 and Ots et al. 2016, where IVOC emissions from diesel were suggested to be a major source of ambient SOA.

We add the references Dunmore et al. 2015 and Ots et al. 2016 in the introduction section when discussing P-S/IVOCs as important precursors to SOA.

We also added the following sentences in the experimental section (at p. 7, L237 in the original text) when describing the way which the IVOC parameters were estimated.

*"In particular, the measured concentrations of speciated and unspeciated IVOCs and their estimated volatility are used to constrain the initial concentration of these species (as discussed in Section 2.2.2 below) as well as to estimate their yields (Zhao et al., 2014). Hodzic et al. (2016) have also estimated the IVOC yields while accounting for wall-losses using recent laboratory studies. However, the yields reported in that study are for a single lumped species, whereas in our work we estimate the yields using 40 IVOC categories, each representing a single compound or a group of compounds of similar structure and volatility. This method allows a more precise representation of IVOC yields and rate constants in the SOA model."*

**R1.5.3** p4, L105-107. If discussing VBS as a conceptual model, the 2-D version (Jimenez et al., 2009)
deserves a mention. Actually, why was this version not used? Box/Lagrangian models have few
restrictions on CPU usage, so would be a natural place to test 2-D VBS schemes.

We address this comment by adding the following paragraph at the end of section 2.2:

*''Simulations of O:C have been previously evaluated in Hayes et al. (2015) using laboratory*
*and field data from CalNex to constrain the predicted O:C. It was concluded in that work that it was*
*not possible to identify one parameterization that performed better than the other parameterizations*
*evaluated, because of the lack of constraints on the different parameters used (e.g. oxidation rate*
*constant, oxygen mass in the initial generation of products and that added in later oxidation*
*generations, SOA yields, and emissions). Therefore, incorporating O:C predictions into the current box*
*model and using those results in the evaluation discussed here would not provide useful additional*
*constraints.''*

We also want to mention that such a discussion would add length to the manuscript, which
might be undesirable as suggested by the reviewers. Not every available parameterization can be tested
in each manuscript, and we have chosen to focus on the 1D VBS parameterizations that are most
commonly used in regional and global models.

**R1.5.4** p4, L118. loses should be losses. (There are other some small typos/English problems throughout,
which should be checked.)

We correct ''loses'' to ''losses'' as suggested and have carefully proofread the revised manuscript.

**R1.5.5** p6, L185-189. The text states that the potential source of error from omission if cold-starts does
not apply to the total amount of vehicular POA emissions. This may be true if the absolute emission
rates are not used, but surely the volatility distribution of cold-start VOC is different to that of warm-
running engines?

We agree with the reviewer that the volatility distribution of POA emissions during cold-starts could be
potentially different from that of warm-running engines, although no information on that comparison
has been published to our knowledge. We have added the sentences below to the text in Section 2.2.2
to clarify this point.

*''It should be noted that the tunnel measurements do not include emissions due to cold starts*
*of vehicles. In the box model, only the relative volatility distribution of vehicular POA measured during*
*the tunnel study is used, and thus this potential source of error does not apply to the total amount of*
*vehicular POA emissions in the model. However, it is still possible that the volatility distribution of POA*
*is different during cold-starts compared to that of POA emitted from warm-running engines. To our*
*knowledge, measurements of the volatility distribution of POA during cold-starts are not available at*
*this time. By comparing the SOA model results using two different POA volatility distributions*
*(Robinson et al. 2007; Worton et al. 2014), we can evaluate to a certain extent the sensitivity of the*
*simulated SOA concentration to the initial POA volatility distribution.''*

**R1.5.6** On the same paragraph though, presumably the Worton et al data could be used to produce a new estimate of total vehicle (S/I)VOC emissions. Why wasn't this done?

It is possible to calculate the P-S/IVOC emissions from the Worton et al. data in the following manner. The emission ratios (in g C L$^{-1}$) for both diesel and gasoline are multiplied by the volume of each fuel sold in Los Angeles county [Gentner et al. Proc. Natl. Acad. Sci. U.S.A. 109, 18318-18323, 2012] to obtain the total emission (in g C) for each of the fuel types. To then implement these total emissions into the box model framework, they are summed and divided by the total CO emissions [Gentner et al. Environ. Sci. Technol. 47, 11837-11848, 2013], which are calculated in a manner analogous to that used for the P-S/IVOCs. A POA/ΔCO ratio of 3 µg m$^{-3}$ ppm$^{-1}$ is obtained, which is lower than the ratio currently used in the box model, 6 µg m$^{-3}$ ppm$^{-1}$ . The difference could be due to a greater influence of diesel emissions at the Pasadena site than is indicated by the fuel sales data or cold starts. These possible sources of error are the reason that the observed POA/ΔCO ratio was chosen for constraining the P-S/IVOC emissions rather than the approach suggested in this comment.

**R1.5.7** p8,L276-279, 281-288. Quantify these time-scales for the reader.

We have modified the text as follows in Section 2.2.

*"Specifically, at lower volatilities (c\* ≤ 1 µg m$^{-3}$), the partitioning kinetics of the organic mass from the particles to the chamber walls have an effective timescale of more than an hour, which is similar or longer than typical chamber experiments (Ye et al., 2016). The limiting step in the partitioning kinetics is evaporation of SVOCs from the particles to the gas phase, and therefore the exact rate of evaporation depends on the OA concentration in the chamber."*

*"According to Krechmer et al. (2016) and other chamber experiments (Matsunaga and Ziemann, 2010), the gas-wall equilibrium timescale doesn't vary strongly with the chamber size. The timescale for gas-wall equilibrium reported in these previous studies was 7 - 13 minutes."*

**R1.5.8** P11, L370 and elsewhere. Define whether mass or volume fractions and stoichiometry are used. This can be an easy error, especially when the cited Donahue paper redefined Raoult's law in terms of mass rather than mole fractions.

We have updated the text as shown below.

*"Where $x_{p,i}$ is the particle phase fraction of lumped species i (expressed as a mass fraction); $C_i$ is the effective saturation concentration, and $C_{OA}$ is the total mass of organic aerosol available for partitioning (µg m$^{-3}$)."*

**R1.5.9** P11, L396. 'shorter' ... than what?

We now specify exactly the photochemical age in the text.

*"The ambient urban SOA mass at the Pasadena ground site is generally measured under conditions corresponding to photochemical ages of 0.5 days or less (Hayes et al., 2013)."*

**R1.5.10** P15, L519-523. Too wordy and repetitive.

We address this comment by updating the text below in the submitted manuscript (at p. 15, L517 in the original text):

**"To make this comparison, the simulated SOA is apportioned between fossil S-SOA, fossil I-SOA, fossil V-SOA, cooking S-SOA, and biogenic V-SOA. The last two apportionments correspond to non-fossil carbon.** *This evaluation is possible following an approach similar to Hayes et al. (2015) where the identity of the precursor is used to apportion SOA."*

**R1.5.11** P15, L531. The work of Dunmore and Ots mentioned above would support this statement.

As suggested, we have added references to Dunmore et al. 2015 and Ots et al. 2016 in the line indicated by the reviewer.

**R1.5.12** P15, L543 on. This paragraph is a good example where the authors attribute all problems to SOA mechanisms. It may well be that the box model setup is responsible for the problems.

We address this comment by updating the conclusions about our results. We kindly refer the reviewer to our updated texts below, which are copied from the updated manuscript.

**From Section 3.1**

*"Finally, the ROB + ZHAO + MA and the WOR + ZHAO + MA cases both better represent SOA formation and exhibit better model/measurement agreement among the different cases used in this work. They are both consistent with the OFR reactor data at longer photochemical ages as shown in Figs. 3 and 4 compared with the other cases. At a qualitative level, the MA parameterization simulations are more consistent with the fit of the OFR measurements in which the SOA mass remains nearly constant at longer photochemical ages. In contrast, the cases with the TSI parameterization do not follow this trend as the SOA mass keeps increasing between 2 and 3 days age, which is not observed in the measurements. As already mentioned, the model used for this work does not include fragmentation reactions, and including these reactions, in particular branching between functionalization and fragmentation during gas-phase SVOC oxidation, may improve the cases using a potential update of the TSI parameterization as discussed below.* **Fig. 4F indicates that including additional P-SVOC mass in the model and accounting for gas-phase wall losses in chamber studies improves SOA mass concentration simulations with respect to the measurements. However, in the WOR + ZHAO + MA case there is still a slight under-prediction of SOA formed at shorter photochemical ages (between 0.05 and 0.5 days), and this discrepancy is observed in all the other model cases.** *Given the uncertainties in the model set-up discussed in the experimental section, it is not possible to conclude if one of the four cases (i.e. ROB + ZHAO + TSI, WOR + ZHAO + TSI, ROB + ZHAO + MA, WOR + ZHAO + MA) more accurately represents SOA formation in the atmosphere."*

We also want to mention that we explain the importance of fragmentation reactions as a response to comment R2.2.1.

**From Conclusions:**

*''Therefore, the model cases with updated VOC yields that account for chamber wall-losses*
*best reproduce the ambient and OFR data. However, while the WOR + ZHAO + MA case appears to*
*represent a slight improvement over the ROB + ZHAO + MA case, as well as over the ROB + ZHAO + TSI*
*and WOR + ZHAO + TSI cases, it is not possible to conclude that one set of parameters is better than*
*the other since the difference in the predictions for these 4 cases (15 % on average) is likely smaller*
*than the uncertainties due to the model setup as well as the lack of a gas-phase fragmentation*
*pathway during aging.''*

**R1.5.13** P17, Sect. 3.2. Given my reservations about the validity of the box-model, and its obvious lack of
treatment of VOC degradation with transport time, I wasn't convinced that this section had a good basis.
In addition, the manuscript is already quite long, and this section feels like a side-issue.

The model includes in fact a detailed treatment of VOC degradation in which the reduction in
concentration with photochemical age is simulated. In fact, the treatment of VOC degradation in the box
model is more rigorous than in 3-D gridded models in that there is no lumping of VOCs and the IVOCs
are speciated, which allows the use of more precise oxidation rate constants. To respond to the concern
regarding manuscript length, we have shortened this section to one paragraph.

**Referee 2 Comments**

**General Comments:**

**R2.1** Ma and coauthors present a follow-up to the work in Hayes et al. (2015), adding more recent parameterizations for I/S-VOC emissions and yields as well as very recent approaches to correcting chamber yields for wall losses. I am satisfied with the application of the box model to the Pasadena data given the lack of quantitative statistics presented. The important limitations of the aging mechanisms and over-exuberant IVOC formation pathways is demonstrated more qualitatively than quantitatively. Also, the authors are careful to avoid strong conclusions about the dominance of SOA from IVOCs over SVOCs or vice-versa. I do urge the authors to move toward a 3D-CTM analysis in the future, particularly since I'm pretty sure input datasets exist for all the major CTMs. Although the conclusions are not exactly novel (other studies have shown that the VBS functionalization mechanisms overpredict at long photochemical lifetimes), I appreciate the demonstration of the improved parameterizations, particularly the chamber wall-loss correction. I found some aspects of the experimental design to have unnecessary limitations. Moreover, I encourage the authors to consider improving several aspects of the presentation before I recommend publication of this manuscript.

We thank the reviewer for their thoughtful comments, and have provided point-by-point responses below. As discussed in response to, e.g., R1.1.1, we disagree with the notion that a 3D-CTM is superior in all cases. In some cases a box model can be complementary and even superior to a 3D-CTM for some applications.

**Specific Comments:**

**R2.2.1** My primary question/criticism is why do the authors not investigate aging mechanisms with fragmentation given the emerging global/regional model implementations of these pathways and the low computational overhead of their own box model? This would seem like an ideal application given the experimental data available to them from CalNex. Why not use measured AMS elemental ratios to help constrain the configuration choices here? Potentially, that analysis could give complementary information to the analysis in Section 3.2 and Fig. 6.

We address to this comment by running two model cases and including a fragmentation process due to heterogeneous oxidation of the particles. The fragmentation is parameterized as an exponential decay of OA concentration with a lifetime of 50 days as reported in Ortega et al. 2016.

[Figure]

Figure 3. Predicted urban SOA mass for the B) ROB + ZHAO + TSI and C) WOR + ZHAO + TSI cases with
the original model set-up for this work.

[Figure]

Figure R1. Predicted urban SOA mass by the A) ROB + ZHAO + TSI and B) WOR + ZHAO + TSI cases when
including fragmentation.

To summarize these findings, we have updated the text in the manuscript and added a discussion of the
different fragmentation mechanisms. We only show results for two cases (those above), but all six
model cases give similar results when including fragmentation.

*"According to the OFR data from Ortega et al. (2016), the mass of OA starts to decay due to*
*fragmentation after heterogeneous oxidation at approximately 10 days of photochemical aging. The*
*results are consistent with other OFR field measurements (George and Abbatt, 2010; Hu et al., 2016;*
*Palm et al., 2016). In this work, the model is run only up to 3 days, which is much shorter than the age*
*when heterogeneous oxidation appears to become important. In fact, when including a fragmentation*
*pathway for each of the model cases, a reduction of OA of only 6 % is observed compared to the cases*
*without fragmentation at 3 days of photochemical aging. In this sensitivity study, the fragmentation is*
*parameterized as an exponential decrease in OA concentration that has a lifetime of 50 days following*

*Ortega et al. (2016). Given the results, the inclusion of fragmentation due to heterogeneous oxidation in the model does not significantly change the model results or the conclusions made in this work.*

*More generally, there are at least three different fragmentation mechanisms that could be responsible for the decrease of SOA formation at very high photochemical ages. The first mechanism is the reaction of oxidants (e.g. OH) with the surface of an aerosol particle and decomposition to form products with higher volatility, i.e. due to the heterogeneous oxidation just described. The second type of fragmentation that may be important for very high photochemical ages in the OFR (Palm et al., 2016) is due to the high concentration of OH. Most of the molecules in the gas phase will react multiple times with the available oxidants before having a chance to condense, which will lead to the formation of smaller products too volatile to form SOA. However, this is only important at very high photochemical ages in the OFR, which are not used in this work. A third type of fragmentation can occur during the aging of gas-phase SVOCs (Shrivastava et al., 2013; 2015). The TSI parameterization used in the model from this work and from previous modeling works (Robinson et al., 2007; Hodzic et al., 2010; Shrivastava et al., 2011) only includes the functionalization of the SVOCs and neglects fragmentation reactions. More recently, Shrivastava et al. (2013) have modified the VBS approach in a box model by incorporating both pathways and performed several sensitivity studies. The results when including fragmentation generally exhibit better agreement with field observations, but as noted in that work the agreement may be fortuitous given that both the emissions as well as the parameters representing oxidation in the model are uncertain. This third type of fragmentation is not simulated in our sensitivity study using the approach above, and it remains poorly characterized due to the complexity of the chemical pathways and the number of compounds contributing to SOA formation as described in Shrivastava et al. (2013).''*

**R2.2.2** The authors discuss extensively the problems with aging VOC oxidation products and the tendency for mechanisms to accumulate mass at long photochemical lifetimes. It is important that they emphasize (stronger than they already do) that this aging approach is likely problematic precisely because it does not consider fragmentation. One of the main conclusions I read from the paper is that wall-loss corrected VOC yields should be used and aging mechanisms turned off. Conceivably, a future study will conclude that turning off aging is a bad idea because even though the OA mass is better predicted at long time, the O:C is underpredicted. The models of the future will hopefully have both more accurate yields and probably aging with both functionalization and fragmentation adequately described. To avoid confusion in the meantime, I recommend the authors refer everywhere to the TSI aging as "aging by functionalization only" or something similar, with an appropriate acronym for readability.

We have addressed this comment in our responses to comments R2.2.1 and R1.5.12. We have not changed the abbreviations in the text to "TSI with aging by functionalization only", since that would be very cumbersome terminology. Instead we very clearly address this important issue in the abstract and the conclusions as well as in new text quoted in our responses to R2.2.1 and R1.5.12.

**R2.2.3** L390-394 and L480-483: The authors repeatedly refer to the OFR work of Ortega et al. (2016) to
justify not including fragmentation in any model case. This argument relies on the assertion that
fragmentation only played a dominant role when the OA mass began to decrease after it had plateaued
for a couple of days in photochemical age space. But the OA concentrations started leveling off in that
study at about 1 day. As with any competition, the manifestation of a plateau indicates to me that
fragmentation is playing a role equal to that of functionalization. So sentences like L477-480 and L482-
483 are pretty confusing, if not misleading.

We completely agree with the reviewer and have modified the text to clarify the manuscript.

At L390 - 394, the new text reads as follows:

*"Since fragmentation and dry deposition are not included in the model, it has only been run to*
*3 days in order to minimize the importance of these processes with respect to SOA concentrations*
*(Ortega et al., 2016). Nevertheless, it is very likely that gas-phase fragmentation of SVOCs (e.g.*
*branching between functionalization and fragmentation) occurs during oxidative aging over these*
*photochemical ages as is discussed in further detail below."*

At L480 - 483, the text has been changed already in response to comment R1.5.12 and can be viewed in
our response to that comment.

**R2.2.4** The application of the wall-loss corrected chamber yields seems problematic to me. First of all,
many of the studies used to inform the Tsimpidi et al. (2010) yield set included seed aerosol in their
experiment. As the authors point out multiple times, the data they have included in Table S4 should be
considered an upper bound. However, I fear that their demonstration of this approach will encourage
others to blanket apply the parameters of Krechmer et al. (2016) to historical chamber yields without
considering the details and possible interferences. I encourage the authors to describe in detail the
problems with applying the narrowly defined Krechmer Cw's to existing data and repeat that paper's call
for more detailed analysis of chamber data before the community gobbles this simple approach and
then moves on to the next hot SOA formation topic.

We are aware of multiple groups that are working on further characterizing vapor wall losses and their
impact on SOA formation experiments. In that context, it seems very unlikely that our simple approach
would become "dominant" in the SOA modeling field. We still believe that it provides one useful
sensitivity study about the impact of the vapor loss problem. We address this comment by adding the
following sentence in the conclusion section of the manuscript.

*''Moreover, uncertainties in the vapor wall-loss corrected yields remain, and the correction of*
*the yields has been performed here using data from a limited number of laboratory studies. In*
*particular, the effect of temperature and humidity on gas-wall partitioning needs to be characterized.*
*The results obtained in our work motivate future studies by showing that SOA models using wall-loss*
*corrected yields reproduce observations for a range of photochemical ages at a level of accuracy that*
*it is as good as or better than parameterizations with the uncorrected yields.''*

**R2.2.5** Why is the SOA mass in Fig. 5 not also divided by CO concentrations to correct for dilution?

As suggested, we have included a right-side axis in the Fig. 5A bar graph representing the SOA mass to
ΔCO concentration ratios.

**R2.2.6** L257-258: Woody et al. (2016) proposed a meat cooking volatility distribution. Why not try this
one in a sensitivity test?

We performed a sensitivity study running the model using the meat cooking volatility distribution
proposed by Woody et al. (2016) as suggested by the reviewer. For ease of comparison, we include here
the original results obtained before the sensitivity study and taken from the submitted manuscript.

[Figure]

Figure 3. Predicted urban SOA mass for the C) WOR + ZHAO + TSI and F) WOR + ZHAO + MA cases with
the original model set-up for this work.

[Figure]

Figure S8. Predicted urban SOA mass for the A) WOR + ZHAO + TSI and B) WOR + ZHAO + MA cases when
using the meat cooking volatility distribution reported in Woody et al. (2016).

We have added the figure above in the supporting information and the paragraph below in the manuscript to summarize the findings.

*"Finally, Woody et al. (2016) recently proposed a meat cooking volatility distribution and therefore we perform a sensitivity study by using this distribution in our model for P-SVOCs coming from cooking sources. The results are displayed in the supporting information (Figure S8), where this alternate approach has been implemented for the WOR + ZHAO + TSI and WOR + ZHAO + MA cases. By comparing the results obtained from this sensitivity study with Fig. 3, the two cases in the sensitivity study display a slight decrease of SOA/ΔCO values over 3 days of photochemical aging with a difference of approximately 8% at 3 days. Thus, the model-measurement comparison does not change significantly relative to the base case. Given the similarities between the sensitivity study and Fig. 3, as well as the possibility of cooking SOA sources other than meat-cooking (i.e. heated cooking oils, Liu et al. (2017)), the remainder of our work uses the Robinson et al. volatility distribution for P-SVOCs from cooking sources."*

**R2.2.7** Can the authors clarify more directly why the model with the Worton parameters for SVOCs gives more OA than that with the Robinson parameters? The volatility distribution and Fig. 2 show pretty clearly that the emissions are substantially higher in volatility. Is the difference really from the added 7.5% mass that comes with the 1-bin aging mechanism? If that's the case, please emphasize more clearly the uncertainty in this parameter/approach to put the differences in these two model runs into context.

We want to clarify that the difference between the results for the Worton and Robinson parameters is not due to the added 7.5% mass during aging but rather the ratio of SVOC/POA at the beginning of the SOA formation. We kindly refer the reviewer to the paragraph below from the submitted manuscript (p. 12, L439). For clarity, we have updated the text.

**"The more rapid SOA formation is due to the updated SVOC volatility distribution in this model case compared to the cases that use the Robinson et al. (2007) distribution.** *Specifically, as shown in Fig. 2F, there is a higher relative concentration of gas phase SVOCs in the $c^* = 10^2$ bin and thus a higher ratio of P-SVOC to POA. Given that in the box model (and in most air quality models) the P-SVOC emissions are determined by scaling the POA emissions according to their volatility distribution, a higher P-SVOC to POA ratio will then result in a higher initial P-SVOCs concentration. Furthermore, SOA formation from P-SVOCs is relatively fast. Together these changes lead to increases in SOA formation during the first hours of photochemical aging when using the Worton et al. volatility distribution."*

**R2.2.8** Section 3.2 and Fig. 6: This analysis is an interesting idea but I don't think the slight differences among the model cases warrant such a long-winded discussion and detailed figure. It would be enough to add a comment to section 3.1 that the WOR cases give more SOA from precursors with kOH in the range identified by Ortega et al. (2016). The abstract and conclusions would need to be correspondingly reduced.

We have shortened the discussion in Section 3.2 as suggested to one paragraph. We have also reduced the relevant paragraph in the conclusions.

**R2.2.9** L716-718 and L796-797: Why do the authors not discuss the limitations of their aging mechanisms that only reduce volatility by one bin at a time? It is possible that a compound can shift more than one generation in volatility upon oxidation; the more recent 2D-VBS approaches and the SOM methods allow for multi-decadal shifts in volatility. Approaches like these might push the products below the "oxidation-partitioning barrier" manifested when compounds are protected from gas-phase oxidation.

We address this comment by kindly referring the reviewer to the paragraph below in the submitted manuscript (p. 20, L712). We have also updated this paragraph for clarity.

> **"With these considerations in mind, the volatility distribution of the SVOCs is somewhat different in the model compared to the measurements. Most notably, the model does not form a significant amount of lower volatility SOA in the $10^{-2}$ µg m$^{-3}$ bin, whereas the measurements have a much higher concentrations in this bin. A factor that may explain this difference between the volatility distributions is the lack of particle phase reactions that continue to transform SOA into lower volatility products, a process which is not considered in the model.** *One example of a particle phase reaction is the formation of SOA within deliquesced particles, including the partitioning of glyoxal to the aqueous phase to produce oligomers as discussed in Ervens and Volkamer (2010), although that specific mechanism was of little significance during CalNex (Washenfelder et al., 2011; Knote et al., 2014). Alternatively, the use of an aging parameterization where the volatility may decrease by more than one order of magnitude per oxidation reaction would also distribute some SOA mass into lower c\* bins. Hayes et al. (2015) previously evaluated different parameters for aging. However, the results from this previous study showed that substantial over-prediction of SOA was observed when using the Grieshop et al. (2009) parameterization in which each oxidation reaction reduced volatility by two orders of magnitude. New parameterizations may be necessary to produce the observed SOA volatility and concentration simultaneously (Cappa et al. 2012). However, we note that the additional low volatility organic mass will not significantly change SOA predictions in urban regions where OA concentrations are relatively high."*

**R2.2.10** L760-762: How do these reaction rate constants compare to estimation methods developed for
the 2D-VBS? If you used those approximations (based on C* and assumed O:C) would you do better?

We are not sure exactly what version of the 2-D VBS the reviewer is referring to, but 2-D VBS
parameterizations have used a single rate constant of $4 \times 10^{-11}$ $cm^3$ $molec^{-1}$ $s^{-1}$ for oxidation and aging of
IVOCs [Murphy et al. Atmos. Chem. Phys. 12, 10797-10816, 2012]. This rate constant is generally higher
than that used in our own work for the initial oxidation reaction, and thus would be expected to
improve the model/measurement agreement at short photochemical age. At the same time, such a
result is not surprising given that the rate constant used in the 2-D VBS was tuned to best match
laboratory and field observations. In contrast, the rate constants from our work are estimated based on
the precursor structure as described in the manuscript as well as in Zhao et al. (2014), and thus they are
better constrained. Furthermore, it should be noted that the aging rate constant used in the box model
for subsequent oxidation reactions is the same as that used in the reference above ($4 \times 10^{-11}$ $cm^3$ $molec^{-1}$
$s^{-1}$).

Furthermore, in the statistical oxidation mode (SOM) [Cappa et al. Atmos. Chem. Phys. 12, 9505-9528,
2012] the reaction rates constants are parameterized using both carbon and oxygen number. When
comparing our rate constants in the supporting information against those summarized in Figure S1 of
the supporting information of Cappa et al., the rate constants are very similar. This result is not
surprising given that both are based on the same structure-activity relationship [Kwok and Atkinson
*Atmos. Environ.* 29, 1685-1695, 1995].

**Minor Changes/Typos:**

**R2.3.1** L48-50: This sentence should say something about how the two methods predict similar mass at
short to moderate photochemical ages.

We address to this comment by updating the sentence below in the manuscript.

‘‘*The model predicts similar SOA mass at short to moderate photochemical ages when the*
*"aging" mechanisms or the updated version of the yields for VOC oxidation are implemented.*’’

**R2.3.2** L82: Consider replacing "nucleate" with "form".

We replace ‘‘nucleate’’ by ‘‘form’’ in the text as suggested by the reviewer.

**R2.3.3** L742-743: Make sure to also mention that Woody et al. (2016) cited excessive model dispersion
as a potential complicating factor.

This is an excellent suggestion and we have updated the sentence below in the manuscript.

*"As stated in Woody et al. (2016), the higher ratio may compensate for other missing (or*
*underrepresented) formation pathways in SOA models or excessive dispersion of SOA in their model."*

**R2.3.4** I recommend adding figures with SOA mass curves (not just the yields) for each of the VOC
product species to the supporting information, thereby visually demonstrating the effects of the upper-
and lower-bound yield parameterizations. It would be a good idea to assume a background
concentration equal to 2.1 ug m$^{-3}$ (or greater if you just want to take an average of your total OA, model
wide) like in the model so that you get relevant partitioning.

We address this comment by adding figures with SOA mass curves for each of the VOC classes in the
supporting information as suggested. These figures are also displayed below.

[Figure]

Figure S1. Predicted urban SOA mass from the alkane VOCs (Alk5) for different SOA formation
parameterizations.

[Figure]

Figure S2. Predicted urban SOA mass from the olefin VOCs (Ole1) for different SOA formation
parameterizations.

[Figure]

Figure S3. Predicted urban SOA mass from the olefin VOCs (Ole2) for different SOA formation
parameterizations.

[Figure]

Figure S4. Predicted urban SOA mass from the aromatic VOCs (Aro1) for different SOA formation
parameterizations.

[Figure]

Figure S5. Predicted urban SOA mass from the aromatic VOCs (Aro2) for different SOA formation
parameterizations.

[Figure]

Figure S6. Predicted urban SOA mass from isoprene (Isop) for different SOA formation
parameterizations.

[Figure]

Figure S7. Predicted urban SOA mass from terpenes (Terp) for different SOA formation
parameterizations.

To summarize these findings, we have added the paragraph below in the manuscript in section 2.2 and
include the Figures S1 to S7 in the supporting information.

*"Furthermore, as described in the supporting information, the updated SOA yields for VOC*
*oxidation result in distribution of SVOC mass into lower volatility bins compared to the original*
*parameterization, although the sum for the SVOC yields ($\alpha_i$) remains similar. In the absence of aging,*
*the SOA yields, Y, resulting from the wall-loss correction should be considered upper limits (MA*
*parameterization), whereas the original yields serve as lower limits due to the considerations*
*discussed above (TSI parameterization without aging). As shown in the supporting information*
*(Figures S1 - S7) when aging (TSI parameterization with aging) is included the SOA yields increase*
*beyond those observed when applying the wall loss correction for most of the VOC classes at longer*
*photochemical ages (it should be noted that SOA masses in Figures S1 - S7 were calculated using the*
*same background as for the other model cases, 2.1 $\mu g\ m^{-3}$). This feature of the aging parameterization*
*is likely to blame for SOA over-predictions observed at long aging times when comparing with*
*ambient data (e.g. Dzepina et al., 2011; Hayes et al., 2015)."*

**R2.3.5** SI, L6-22: I found the derivation of the wall-loss correction confusing. First, the quantity in
parentheses in equations 5 and 6 should be the reciprocal. I assume they used the correct form for the
calculation because I calculated the adjusted ARO2 and it would have been way off using the equation
as it is written. Also, [VOC] should be replaced with something more accurate like [ΔROG] or [ΔVOC]. It
would be helpful to explain briefly why the mass of compounds on the walls, Cw, is a function of C*
alone and not Ctot, Cg or Cp. This is essentially a consequence of the equilibrium assumption in the
chamber analysis, as I understand it.

We have corrected equations 5 and 6. We also replace [VOC] by [ΔVOC] as suggested. We also address
this comment by clarifying that $c_w$ is the equivalent organic mass concentration of the walls. The
notation $c_w$ is used by Krechmer et al. (2016) and we have kept it for consistency. We add the text below
in the supporting information for clarity.

*"For clarity, $c_w$ is the equivalent organic mass concentration of the walls, and it is an*
*empirically determined value. Equations 2 and 3 are the partitioning equations that describe either*
*the partitioning between the gas phase and walls or the gas phase and the particles, which both*
*depend on the volatility of the organic vapors, $c^*$. The significance of $c_w$ can be understood by*
*comparing equations 2 and 3. In equation 3, the partitioning is dependent on the total particle phase,*
*$c_{OA}$. Similarly, the parameter $c_w$ is the amount of mass in the chamber walls available for partitioning*
*expressed as an effective mass concentration based on the work of Krechmer et al. (2016). However,*
*the value of $c_w$ is a function of $c^*$ as shown in equation 1."*

[revised manuscript text omitted]
_{OA}} + \frac{C_{w,1}}{C_{OA}}\right)^{-1} + \alpha_{10} \left(1 + \frac{10}{C_{OA}} + \frac{C_{w,10}}{C_{OA}}\right)^{-1} + \alpha_{100} \left(1 + \frac{100}{C_{OA}} + \frac{C_{w,100}}{C_{OA}}\right)^{-1}$$

$$+ \alpha_{1000} \left(1 + \frac{1000}{C_{OA}} + \frac{C_{w,100}}{C_{OA}}\right)^{-1} \tag{6}$$

The corrected yields in this work were determined by simulating yield curves using the parameters published in Tsimpidi et al. (2010) and then refitting the curves using Equation 6.

For clarity, $c_w$ is the equivalent organic mass concentration of the walls, and it is an empirically determined value. Equations 2 and 3 are the partitioning equations that describe either the partitioning between the gas phase and walls or the gas phase and the particles, which both depend on the volatility of the organic vapors, $c^*$. The significance of $c_w$ can be understood by comparing equations 2 and 3. In equation 3, the partitioning is dependent on the total particle phase, $c_{OA}$. Similarly, the parameter $c_w$ is the amount of mass in the chamber walls available for partitioning expressed as an effective mass concentration based on the work of Krechmer et al. (2016). However, the value of $c_w$ is a function of $c^*$ as shown in 
[revised manuscript text omitted]

[Figure]

**Figure S1.** Predicted urban SOA mass from the alkane VOCs (Alk5) for different SOA formation parameterizations.

[Figure]

**Figure S2.** Predicted urban SOA mass from the olefin VOCs (Ole1) for different SOA formation parameterizations.

[Figure]

**Figure S3** Predicted urban SOA mass from the olefin VOCs (Ole2) for different SOA formation parameterizations.

[Figure]

**Figure S4.** Predicted urban SOA mass from the aromatic VOCs (Aro1) for different SOA formation parameterizations.

[Figure]

**Figure S5.** Predicted urban SOA mass from the aromatic VOCs (Aro2) for different SOA formation parameterizations.

[Figure]

**Figure S6.** Predicted urban SOA mass from isoprene (Isop) for different SOA formation parameterizations.

[Figure]

**Figure S7.** Predicted urban SOA mass from terpenes (Terp) for different SOA formation parameterizations.

[Figure]

**Figure S8.** Predicted urban SOA mass by the A) WOR + ZHAO + TSI and B) WOR + ZHAO + MA cases when using the meat cooking volatility distribution reported by Woody et al. (2016).

[Figure]

**Figure S9.** Estimated fractional contributions to urban SOA mass concentration using the WOR + ZHAO + MA case.

[Figure]

**Figure S10.** Predicted urban SOA mass for the A) ROB + ZHAO + MA and B) WOR + ZHAO + MA cases when using IVOC initial concentrations determined using photochemical age, the Pasadena IVOC concentrations and the estimated IVOC oxidation rate constants.

[Figure]

**Figure S11.** Organic mass fraction remaining as a function of temperature for Pasadena, California during CalNex 2010. Data correspond to 12:00 – 15:00 local time.

---

## Author Response (AR2)

**Comment from Editor and author's response (Ma et al.)**

**Editor:** I'd like to thank the authors for their thorough consideration of the reviewers' comments, their responses and revised manuscript. Prior to final acceptance, I'd like to invite the authors to respond to the last remaining point of referee #1: "Presumably it is the curve fitting routine, applied with Eqn 6 from the Supplement, which produces the new alphas, I wonder though if the same fit routine, applied without wall-loss, would reproduce the same alpha's as the original Tsimpidi? Assuming that this is the case, then I would just ask the authors to add some words on these large changes, and implications for simulations of regional scale OA concentrations". This is potentially important and worth commenting on. I am happy to consider this as a minor revision without sending for re-review. I congratulate the authors on a thorough and well-considered manuscript, and an enjoyable read.

**Response:** We thank the editor and the reviewers again for their feedback. We have verified that indeed if the curve fitting routine is applied without wall-losses, then the original Tsimpidi alpha values are obtained. In addition, a paragraph has been added to the end of Section 3.1 to discuss the implications for simulations of regional scale SOA concentrations. This is paragraph is quoted below (in bold) for the editor's convenience.

[revised manuscript text omitted]